# Arginine GlcNAcylation of Rab small GTPases by the pathogen *Salmonella* Typhimurium

Kun Meng[1,2,3,7], Xiaohui Zhuang[2,3,7], Ting Peng[2,3,7], Shufan Hu[2,3], Jin Yang[1], Zhen Wang[4], Jiaqi Fu[4], Juan Xue[1], Xing Pan[1,2,3], Jun Lv[1], Xiaoyun Liu [5], Feng Shao [6] & Shan Li [1,2,3✉]

*Salmonella enterica* serovar Typhimurium, an intracellular Gram-negative bacterial pathogen, employs two type III secretion systems to deliver virulence effector proteins to host cells. One such effector, SseK3, is a Golgi-targeting arginine GlcNAc transferase. Here, we show that SseK3 colocalizes with *cis*-Golgi via lipid binding. Arg-GlcNAc-omics profiling reveals that SseK3 modifies Rab1 and some phylogenetically related Rab GTPases. These modifications are dependent on C-termini of Rabs but independent of the GTP- or GDP-bound forms. Arginine GlcNAcylation occurs in the switch II region and the third α-helix and severely disturbs the function of Rab1. The arginine GlcNAc transferase activity of SseK3 is required for the replication of *Salmonella* in RAW264.7 macrophages and bacterial virulence in the mouse model of *Salmonella* infection. Therefore, this SseK3 mechanism of action represents a new understanding of the strategy adopted by *Salmonella* to target host trafficking systems.

[1] Institute of Infection and Immunity, Taihe Hospital, Hubei University of Medicine, Shiyan, Hubei, China. [2] College of Life Science and Technology, Huazhong Agricultural University, Wuhan, Hubei, China. [3] College of Biomedicine and Health, Huazhong Agricultural University, Wuhan, Hubei, China. [4] Institute of Analytical Chemistry, College of Chemistry and Molecular Engineering, Peking University, Beijing, China. [5] Department of Microbiology, School of Basic Medical Sciences, Peking University Health Science Center, Beijing, China. [6] National Institute of Biological Sciences, Beijing, China. [7] These authors contributed equally: Kun Meng, Xiaohui Zhuang, Ting Peng. ✉email: lishan@mail.hzau.edu.cn

Rab GTPases are the largest branch of the Ras superfamily of small GTPases, representing over 60 members in humans[1,2]. They are critical regulators of endocytic and secretory membrane-trafficking events and their activity is tightly controlled within cells[3]. Rabs are targeted to membrane compartments, a process that depends on prenylation, a kind of posttranslational modification with one or two geranylgeranyl groups on their C-terminal cysteine residue(s)[4]. Rab GTPases switch between two distinct conformational states, the active GTP-bound form and the inactive GDP-bound form, regulated by guanine nucleotide exchange factors (GEFs) and GTPase-activating proteins (GAPs). In the GTP-bound form, Rab GTPases recruit and activate specific sets of effector proteins, thus regulating membrane traffic events[2,5]. Upon GTP hydrolysis, GDP-bound Rabs are retrieved from the membrane by Rab GDP-dissociation inhibitor (GDI) for a new cycle[6]. Manipulation of Rab GTPase function is often used by intracellular bacterial pathogens to facilitate entry, replication, and intracellular survival[7].

*Salmonella enterica* serovar Typhimurium (*S.* Typhimurium) is a kind of Gram-negative facultative intracellular bacterial pathogen, capable of causing both localized and systemic disease in a broad range of hosts[8]. *S.* Typhimurium pathogenesis is highly dependent on two type III secretion systems (T3SS) that are encoded within the pathogenicity islands 1 (SPI-1) and 2 (SPI-2), which function in the transport of bacterial effector proteins to the cytoplasm of the host cell[9]. SseK1, SseK2, and SseK3 are SPI-2 effectors that share high sequence identities with the enteropathogenic *Escherichia coli* T3SS effector NleB1[10,11]. Based on the knowledge of NleB1[12,13], other groups and us report that SseK1 and SseK3 show arginine GlcNAc transferase activity and modify the death domain of tumor necrotic factor receptor-1 (TNFR1)-associated death domain protein (TRADD) and TNFR1, respectively[14–16]. However, a phenomenon exists whereby both the SseK3 protein and arginine-GlcNAcylated protein localize on the Golgi apparatus during *Salmonella* infection[17]. Thus, the molecular mechanism and host targets of SseK3, as well as their roles in *Salmonella* infection, remain largely unknown.

Here we show that SseK3 colocalizes with *cis*-Golgi via lipid binding, and ectopic expression of SseK3 leads to Golgi fragmentation. Furthermore, SseK3, but not SseK1, can modify a set of Rab GTPases, including Rab1 and some phylogenetically related Rab proteins, through arginine GlcNAcylation. This modification is dependent on the C-terminal prenylation of Rab1 but independent of the GTP- or GDP-bound forms. We show that SseK3 modifies Rab1 within the switch II region and the third α-helix, and that modification severely disturbs the GTPase activity, binding with GDIs, as well as the membrane cycle of Rab1. SseK3 inactivates Rab1 and disrupts ER-to-Golgi trafficking through arginine GlcNAcylation. SseK3 blocks the host inflammatory cytokine secretion during *Salmonella* infection and is crucial for bacterial virulence in mice.

## Results

### Salmonella-infection-delivered SseK3 localizes on the *cis*-Golgi network.
Previously, translocation of SseK1 and SseK3 during *Salmonella* infection into RAW264.7 macrophage cells was shown[17]. To further analyze the dynamic process of translocation, the C terminus of SseK effectors were fused with a SunTag, which can recruit up to 24 copies of green fluorescent protein (GFP), thereby enabling signal amplification and long-term imaging of a single protein by fluorescence microscopy[18]. SseK1 and SseK1 enzymatic dead mutant (SseK1 DxD) diffused in the cytoplasm with no specific subcellular localization. In contrast, translocated SseK3 began to form a punctate perinuclear structure at 6 h post infection (Fig. 1a and Supplementary Fig. 1) and showed a clear

colocalization with the host Golgi network (labeled with anti-GM130 antibody) (Supplementary Fig. 2a and Supplementary Movie 1–3). Interestingly, the SseK3 enzymatic dead mutant, SseK3 DxD, formed the puncta with a lower speed than the wild-type (WT) SseK3 (Supplementary Fig. 1).

Next, we studied the localization of Arginine-GlcNAcylated proteins catalyzed by the SseK effectors during infection. Arginine-GlcNAcylation signals were completely absent in cells infected with the Δ*sseK1/2/3* mutant complemented with vector and SseK3 DxD. However, the Δ*sseK1/2/3* mutant expressing SseK1 or SseK3 gave rise to an Arginine-GlcNAcylation signal in the host cell cytoplasm (SseK1) or both the cytoplasm and at the Golgi network (SseK3), respectively (Supplementary Fig. 2b). The different subcellular localization of Arginine-GlcNAcylated proteins suggests that translocated SseK1 and SseK3 may target a distinct subset of host substrates during infection.

Nocodazole is a microtubule-depolymerizing drug that is well known to break down the Golgi ribbon into dispersed ministacks[19]. Nocodazole-induced ministacks can be discriminated between *cis*- and *trans*-Golgi cisternae by light microscopy[20]. As expected, *cis*- and *trans*-Golgi (indicated by GM130 and p230, respectively) were separated in nocodazole-treated cells, as judged by the lack of overlay and the low Pearson's coefficient value (Supplementary Fig. 3a). To determine the predominant intra-Golgi localization of SseK3, HeLa cells were infected with *S.* Typhimurium Δ*sseK1/2/3* complemented with a plasmid expressing SseK3 and were treated with nocodazole. The ministacks of SseK3 exhibited prominent colocalization with endogenous GM130 rather than p230 (Fig. 1b). Similar results were obtained for the nocodazole-treated SseK3-transfected cells (Supplementary Fig. 3b). Therefore, SseK3 shows colocalization with *cis*-Golgi structures.

### SseK3 localizes on the Golgi apparatus via lipid binding.
The mechanisms and signals of distribution of glycosyltransferases across cisternae of the Golgi are poorly understood and may vary from one enzyme to another[21,22]. The reason why SseK3 locates on Golgi apparatus is intriguing. Recent studies have shown that polybasic regions promote the association of proteins within a particular lipid environment of the organelle[23,24]. To test whether the polybasic region in SseK3 mediates its recruitment through a similar mechanism, we performed a polybasic region screen in SseK3. We identified and mutated five polybasic regions within the SseK3 structure (Supplementary Fig. 4a). The mutation K87A/R89A/K234A abolished the SseK3 Golgi localization (Fig. 1c, d), unlike other mutants that exhibited a similar Golgi localization with WT SseK3 (Fig. 1c, d and Supplementary Fig. 4b). We found that SseK3 bound to several negatively charged phospholipids by lipid blot assay and the K87A/R89A/K234A mutant showed a reduced binding affinity with PtdIns(4,5)P$_2$ (Fig. 1e), indicating that the Golgi localization of SseK3 may rely on binding to PtdIns(4,5)P$_2$. To further verify this, we took advantage of an inducible recruitment system of phospholipid phosphatases[25]. In this system, the addition of rapamycin promotes the heterodimerization of FKBP12 (FK506-binding protein, 12 kDa) and FRB (FKBP12-rapamycin-binding domain), thereby recruiting the phosphatase to the Golgi, where it hydrolyzes its target phospholipid (Supplementary Fig. 5a). Expression of the PtdIns(4,5)P$_2$ phosphatase, PIP4P1, decreased the Golgi localization of SseK3, whereas the enzymatically deficient mutant had no effects on SseK3 localization (Fig. 1f and Supplementary Fig. 5b). Thus, binding to PtdIns(4,5)P$_2$ is crucial for SseK3 to localize on the Golgi apparatus.

### Ectopic expression of SseK3 disrupts Golgi structure in mammalian cells.
Many intracellular bacterial T3SS effectors can

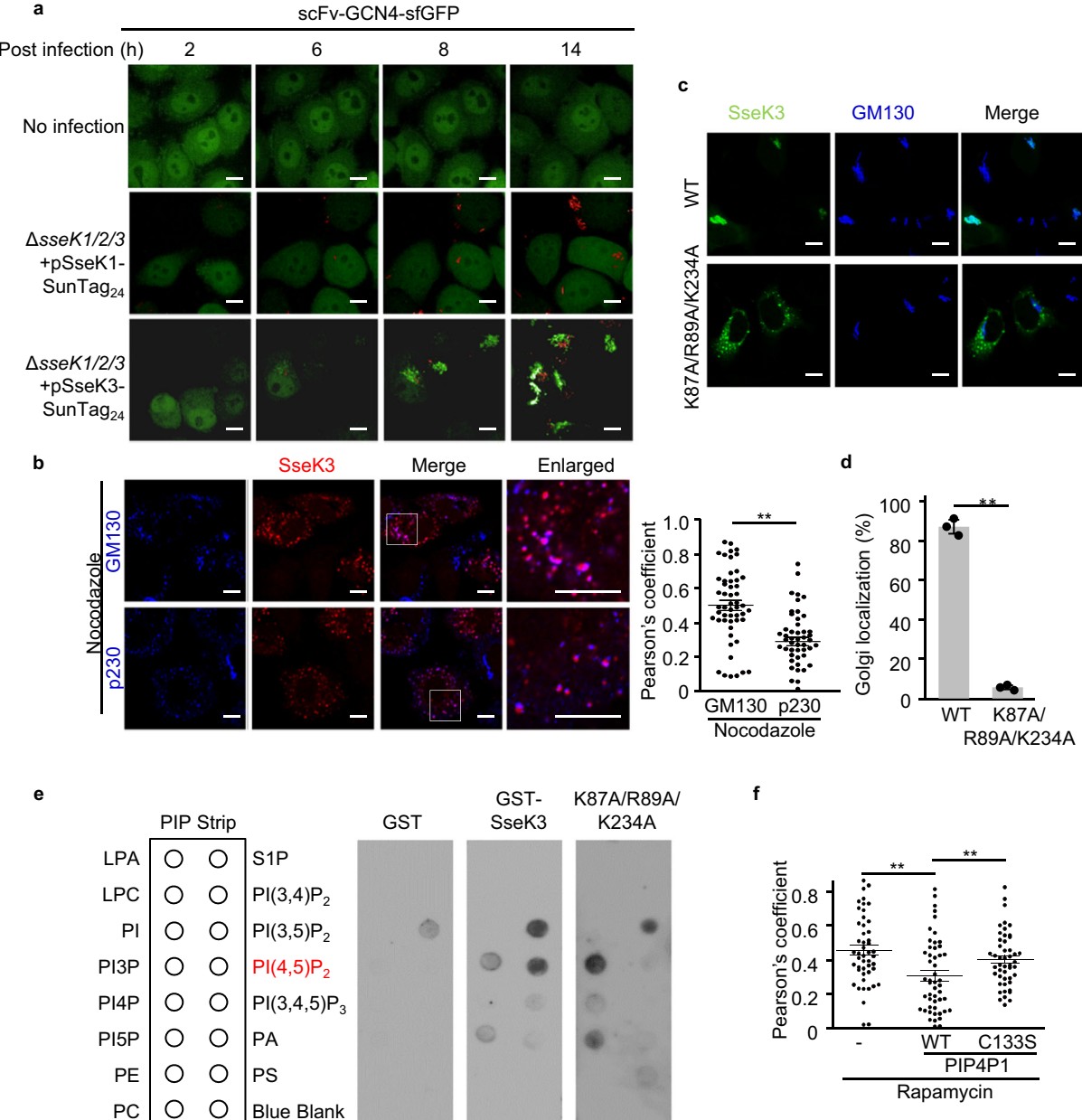

**Fig. 1 SseK3 localizes on *cis*-Golgi apparatus via binding to PtdIns (4,5)P₂. a** Subcellular localization of T3SS-translocated SseK1 and SseK3. HeLa cells stably expressing scFv-GCN4-GFP were infected with *S.* Typhimurium Δ*sseK1/2/3* complemented with a plasmid expressing SseK1-SunTag₂₄ or SseK3-SunTag₂₄. Mock (no infection) cells were set as a negative control. Shown are fluorescence images taken at the indicated time post infection. **b** *Cis*-Golgi localization of SseK3 during infection. HeLa cells were infected with *S.* Typhimurium Δ*sseK1/2/3* complemented with a plasmid expressing SseK3-Flag, treated with Nocodazole for 1 h, and then subjected to immunofluorescence staining using anti-Flag antibody together with an anti-GM130 or anti-p230 antibody. Colocalization of SseK3 with GM130 or p230 are shown in fluorescence images (left) and the statistics of Pearson correlation coefficient (right). The Pearson correlation coefficient was calculated from more than 50 ministacks for each experiment by applying the ImageJ software (http://rsb.info.nih.gov/ij/). Vertical lines represent SEM. **P < 0.01. Data are representative of three independent experiments. Scale bar, 10 μm. **c, d** Effects of polybasic regions (PBR) on Golgi localization of transfected SseK3. **c** HeLa cells were transfected with a plasmid expressing GFP-tagged WT SseK3 or its PBR-mutant. Shown are immunofluorescence staining using anti-GM130 antibody (Scale bar, 10 μm). **d** Statistics of cells are listed according to **c** and the percentages of Golgi localization of SseK3 are mean ± SD from three determinations. **P < 0.01. **e** Lipid binding of the wild-type SseK3 and the K87A/R89A/K234A mutant. **f** Effects of phosphatase on the Golgi localization of SseK3. Statistics of Pearson correlation coefficients showing Golgi localization are listed according to Supplementary Fig. 5. Vertical lines represent SEM. **P < 0.01.

alter host vesicle trafficking when expressed in mammalian cells, a property that is meaningful in defining the functional mechanism of these effectors. Prompted by this notion, we examined several intracellular organelles in cells expressing the family of SseK effectors. Overexpression of SseK3 in HeLa cells caused a severe disruption of the *cis*- and *medial*-Golgi structures stained by the

anti-GM130 and anti-ManII antibodies, respectively (Supplementary Fig. 6a, b). Ectopic expression of SseK3 also disrupted the Golgi structure in 293T cells (Supplementary Fig. 7a). Furthermore, *Salmonella*-infection-delivered SseK3 disrupted Golgi structures 24 h post infection (Supplementary Fig. 8a, b). These disruptions were in a SseK3 dose-dependent manner (Supplementary Figs. 9 and 10).

This outcome of Golgi structure disruption resembled that of the T3SS effector VirA from *Shigella flexneri*, a GAP of Rab1[26]. The phenomenon was dependent on the arginine GlcNAc transferase activity of SseK3, but not SseK1, which suggests that SseK3 may have other enzymatic substrates related to the structure and function of Golgi apparatus (Supplementary Fig. 6a, b). The prominent Golgi disrupting activity of SseK3 was further confirmed by using other Golgi markers, including Golgi structure proteins (GFP-golgin84 and GFP-p230/golgin-245), Golgi-located GTPase proteins (Rab1A, Rab33B, and Rab6A), and Golgi-located glycosyltransferases (GalT-GFP and GalNAc-T2-GFP) (Supplementary Fig. 6c, d). However, SseK3 had no effects on the morphology of the ER, endosome, or lysosome, which were assayed by GFP-Sec22B transfection, anti-EEA1, or anti-LAMP1 immunostaining, respectively (Supplementary Fig. 6c, d). Thus, these analyses unveil that SseK3 plays a functional role at the Golgi apparatus and suggest substrate specificity of SseK3.

**SseK3 catalyzes arginine GlcNAcylation on Rab GTPases during *Salmonella* infection.** Although several possible host substrates of SseK3 have been reported, most of these studies focus on the death domain-containing proteins and are based on the previous knowledge of NleB1[14,17,27,28]. To identify new host substrates, we enriched arginine-GlcNAcylated proteins with indicated antibodies under transfection and infection conditions, and subjected them to immunoprecipitation-mass spectrometry (IP-MS) analyses (Supplementary Fig. 11a). After immunoaffinity enrichment with the Arg-GlcNAc antibody, a range of substrates was detected by silver staining and immunoblotting analyses (Supplementary Fig. 11b). MS was performed to compare the triple-deletion mutant Δ*sseK123* complemented with SseK3 or mock vector. The ratio was calculated as spectral counts in SseK3-proficient samples divided by those in SseK3-deficient controls. We performed transfection and bacterial infection in 293T cells and HeLa cells, respectively, and then selected the top 10% of modified targets by MS analyses as above. Surprisingly, eight out of ten common targets were Rab GTPases between transfection and infection samples (Fig. 2a). During the *Salmonella* infection, more than 20 Rab GTPases corresponding to SseK3 delivery were detected by IP-MS (Fig. 2b). To exclude the effect of endogenous protein abundance on the IP-MS analyses and search for the general substrate(s) of SseK3, we further performed the comparative IP-MS analyses in both mouse embryonic fibroblast (MEF cells) and immortalized bone-marrow-derived macrophage (iBMDM cells). Aside from some highly abundant ribosome proteins, Rab1A and Rab1B were common substrates in the three infected cell types (Fig. 2c). Thus, Rab GTPases are potential host targets of SseK3 during *Salmonella* infection.

To determine which Rab is the bona fide target of SseK3 during *Salmonella* infection, a panel of 35 mammalian small GTPases were analyzed. We found that SseK3 preferred to modifying Rab family GTPases rather than Rho or Ras family GTPases. Furthermore, SseK3 showed a prominent activity toward Rab1. Several other Rabs displaying close phylogenetic relationship to Rab1, including Rab8, Rab18, Rab35, Rab33, Rab30, Rab19, and Rab37, could also be modified by SseK3 upon infection (Fig. 2d and Supplementary Fig. 12). Accordingly, endogenous Rab1 but not Rab5 showed a clear and well-defined colocalization with the arginine GlcNAcylation catalyzed by SseK3 during *Salmonella* infection (Supplementary Fig. 13a, b). This result indicates that Rab1 is the preferred substrate in vivo. Besides, MS analysis identified a 203 Da mass increase on Rab1 upon infection with the SseK3-proficient strain, whereas no mass increase was detected with the dead enzyme SseK3 DxD-proficient strain (Fig. 2e). This 203 Da mass increase indicated a GlcNAc modification.

Furthermore, SseK3 could specifically modify Rab GTPases but not the death domain of TRADD, which was the host target of SseK1 and the EPEC T3SS effector NleB (Fig. 2f). These results showed the differential substrate specificity of these effectors. Thus, SseK3, but not SseK1 or SseK2, specifically modifies Rab GTPases, especially Rab1, during *Salmonella* infection.

**Silencing Rab prenylation signals abolishes the arginine GlcNAcylation by SseK3.** Rab proteins switch between an active GTP-bound state and an inactive GDP-bound state[3]. Meanwhile, as a prerequisite for their function and localization to internal membranes, Rab proteins need to be prenylated at C-terminal cysteine residues in eukaryotic cells[4]. To determine whether SseK3 has a preference for Rab1 in one of these three forms, we performed three tests. First, when Rab1 proteins were co-expressed with SseK3 in both prokaryotic cells and eukaryotic cells, SseK3 exhibited a markedly higher enzymatic activity towards Rab1 in 293T cells than that in *E. coli* (Figs. 2e and 3a, and Supplementary Fig. 14). This indicates that some process occurred in eukaryotic cells may facilitate the arginine GlcNAcylation on Rab1 by SseK3. Second, transfected and *Salmonella*-infection-delivered SseK3 could efficiently modify Rab1 WT, Rab1 S25N (GDP-locked form), as well as Rab1 Q70L (GTP-locked form) to the same level. However, silencing of the prenylation signal by deletion or mutation of the two terminal cysteine residues of Rab1 abolished the modification in both transfection and infection systems (Fig. 3b, c). Third, in an in-vitro reaction system, SseK3 efficiently modified the Rab1 purified from 293T cells but not the Rab1 purified from *E. coli* (Fig. 3d). These data suggest that the prenylation and other processes occurred on Rab1 in the eukaryotic cells are important for SseK3 modification. Consistently, silencing the prenylation signals by deletion of the C-terminal cysteine residues of Rab8 and Rab33 dampened the SseK3 modification during bacterial infection (Fig. 3e). Thus, the C-terminal prenylation signal is crucial for the modification of Rab proteins by SseK3 both in vitro and in vivo.

**SseK3 modifies Rab1 on Arg72, Arg74, Arg82, and Arg111.** To pinpoint the modification site(s) on Rab1, we affinity-purified Flag-tagged Rab1 from 293T cells infected by a *sseK1/2/3* triple-deletion *Salmonella* mutant strain complemented with SseK3 WT or its catalytically inactive mutant SseK3 DxD, respectively. Upon tandem MS (MS/MS) analysis, we detected four chymotryptic and tryptic digested peptides with a mass shift of 203 Da in the SseK3 but not SseK3 DxD complementation group (Supplementary Figs. 15 and 16). The 203 Da increase corresponds to the attachment of one GlcNAc molecule, suggesting multiple modification sites. By MS/MS analyses, these modification sites were mapped to Arg72, Arg74, Arg82, and Arg111 (Fig. 4a and Supplementary Figs. 17 and 18). Modifications were further confirmed by using the mutants Rab1 R72K/R74K/R82K/R111K (R4K) and Rab1 R72A/R74A/R82A/R111A (R4A). Immunoblotting analyses showed that Rab1 R4K (Fig. 4b) and R4A (Supplementary Fig. 19), compared with Rab1 WT, exhibited a vanished modification by SseK3 during *Salmonella* infection. Among these modification sites, Arg72 and Arg82 located within the Switch II region, are conserved in SseK3 modified-Rab GTPase proteins (Fig. 4c). R72A/R82A and R72A/R74A/R82A/R111A (R4A) mutants led to an attenuation in GAP-stimulated GTPase hydrolysis and in response to DrrA, a bacteria-derived GEF in the GEF activity assay (Fig. 4d, e).

**SseK3 inactivates Rab1 through arginine GlcNAcylation and disrupts ER-to-Golgi trafficking.** Next, we investigated the effects of arginine GlcNAcylation on the function of Rab1. Both modified

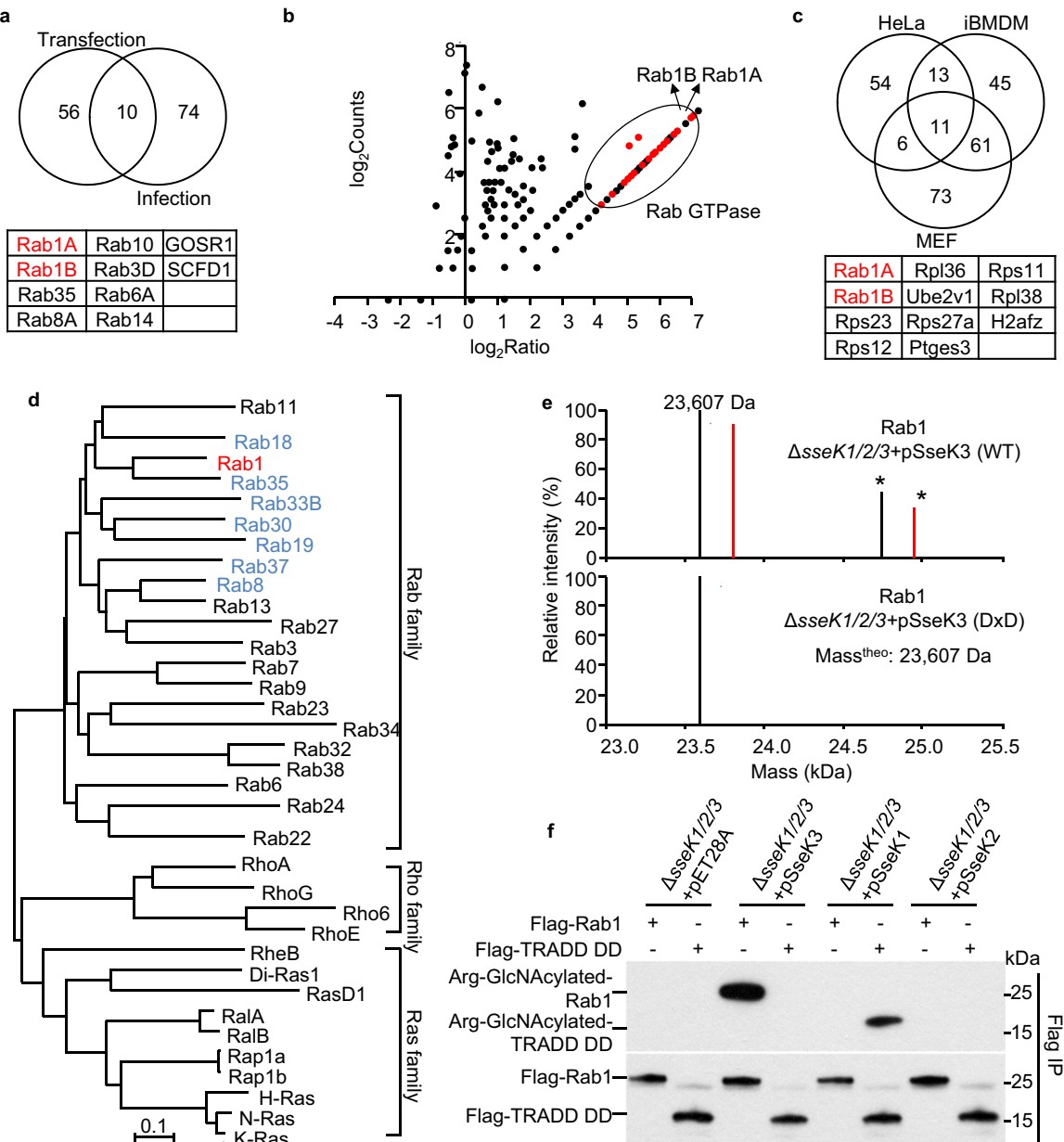

**Fig. 2 SseK3 GlcNAcylates Rab small GTPases during *S.* Typhimurium infection. a** Overlap of Arg-GlcNAcylated proteins identified in 293T cells treated with transfected- or *Salmonella*-delivered SseK3 expression. Protein names in common are listed below. **b** Scatter plots of protein ratios as a function of their relative abundance. Proteins were immunoprecipitated with anti-Arg-GlcNAc antibody and subjected to LC-MS/MS analysis. The ratio was calculated as spectral counts in *S.* Typhimurium Δ*sseK1/2/3*-pSseK3-infected samples divided by those in Δ*sseK1/2/3*-pVec-infected samples. Large ratios indicate preferential detection and modification in HeLa cells infected with SseK3-proficient strains. Red dots correspond to modified Rab GTPase proteins. **c** Overlap of Arg-GlcNAcylated proteins identified in *Salmonella* infected HeLa, MEF, and iBMDM cells. Protein names in common are listed below. **d** Modification of 35 small GTPase proteins by *Salmonella*-infection-delivered SseK3 was tested and shown in Supplementary Fig. 12. Shown here is an unrooted phylogenetic tree computed from the amino acid sequences of these tested small GTPase proteins. The analysis was performed by neighbor-joining in MEGA 5.0 software. The scale bar indicates an evolutionary distance of 0.1 aa substitution per position in the sequence. The GlcNAcylated GTPases are shown in red (strong modification) and blue (moderate modification). **e** Electrospray ionization mass spectrometry (ESI-MS) determination of the total mass of Rab1 immunopurified from *Salmonella*-infected 293T cells. Δ*sseK1/2/3*, SseK1, SseK2, and SseK3 triple-deletion strain of *S.* Typhimurium SL1344 strain. pSseK3 and pSseK3 DxD, SseK3 and SseK3 DxD (D226A/D228A) rescue plasmid, respectively. Modified Rab1 is shown in red. Asterisk symbols denote the prenylated forms of Rab1. **f** SseK3 and SseK1 modifiy Rab1 and death domain of TRADD during *S.* Typhimurium infection, respectively. 293T cells were transfected with Flag-TRADD DD or Flag-Rab1, and then subjected to *Salmonella* infection. Lysates were immunoprecipitated with anti-Flag antibody and immunoblotted with indicated antibodies.

and unmodified versions of Rab1 were purified and incubated with GTP in reactions with or without the bacterial protein LepB, a Rab1 GAP[29]. As expected, the Rab1 GTPase activity was elevated in the presence of GAP. However, a modification on Rab1

inhibited the GTPase activity, both in the presence and absence of GAP (Fig. 5a and Supplementary Fig. 20b). The modification of Rab1 also disturbed the GEF activity of DrrA (Fig. 5b and Supplementary Fig. 20c). The inhibition effects of Rab1 GTPase

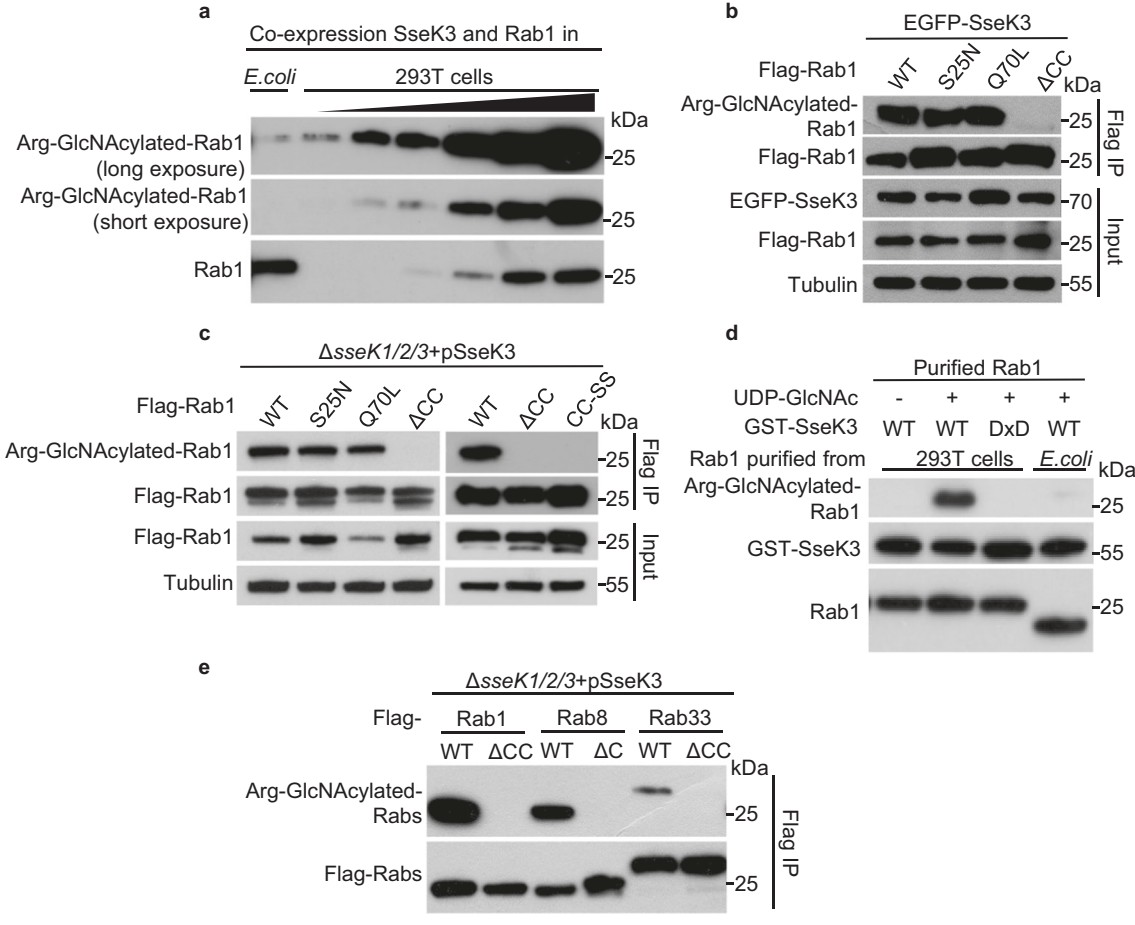

**Fig. 3 Silencing prenylation signals of Rab small GTPases abolishes the GlcNAcylation by SseK3. a** Arg-GlcNAcylation detection of Rab1 purified from prokaryotic and eukaryotic cells. Anti-Rab1 is shown as loading control. **b**, **c** Effects of GTP loading (Q70L)-, GDP loading (S25N)-, and loss of prenylation (ΔCC and CC-SS) forms of Rab1 on the GlcNAcylation by SseK3 during ectopic expression (**b**) and *Salmonella* infection (**c**). Lysates were immunoprecipitated with Flag antibody and immunoblotted with indicated antibodies. **d** Effects of prenylation of Rab1 in a recombinant reaction in vitro. Rab1 with or without prenylation indicates the Rab1 protein purified from 293T cells or *E. coli* BL21 (DE3) strain, respectively. **e** Prenylation of some Rab small GTPases is crucial for the GlcNAcylation by SseK3 during *Salmonella* infection. 293T cells were transfected with Flag-tagged Rab1, Rab8, Rab33, as well as their C-terminal deletion forms, and then subjected to pathogen infection with indicated strains. Shown are immunoblots of anti-Flag immunoprecipitates (Flag IP). Representative data from at least three repetitions are shown.

activity and GEF activity were promoted by enrichment of the modified version of Rab1 (Supplementary Fig. 20).

The switch I and II regions of Rab proteins are the determinants for interactions with their binding proteins and disturbing these regions may reduce the binding affinity[5]. Rab1 interactome analysis showed that the expression of SseK3 but not vector (Fig. 5c) or SseK3 DxD mutant (Supplementary Fig. 21) inhibited the binding of Rab1 with GDI-1 (GDI-α) and GDI-2 (GDI-β). These results were supported by two additional pieces of evidence. First, immunoblotting analyses showed that SseK3 expression abolished the interaction of Rab1 and endogenous GDI-1 and GDI-2, but not other Rab1 regulatory proteins, including p115 (Fig. 5d) or GM130 (Supplementary Fig. 22). Second, recombinant GST-GDI-1 and GST-GDI-2 could readily pull down unmodified-Rab1 but not modified-Rab1 in vitro (Fig. 5e). Thus, SseK3-catalyzed arginine GlcNAcylation on Rab1 blocks the GTPase activity, GEF activity, as well as the interaction with GDI proteins. These results suggest that SseK3 may modulate Rab1 cycling onto and off of the membrane. Consistent with this notion, co-expression of SseK3, but not the SseK3 DxD mutant, led to Rab1 partitioning into the membrane phase (Fig. 5f, g). Thus, SseK3 inhibits the GDI association with Rab1, rendering it more likely to retain on the membrane structures.

Rab1 is required for ER-to-Golgi anterograde transport and inactivation of Rab1 is anticipated to inhibit the exocytic pathway[30]. This idea was confirmed by the observation that SseK3 expression inhibited the secretion of human growth hormone (hGH) to a similar degree compared with LepB_NTD. In contrast, NleB, SseK1, SseK2, and SseK3 DxD mutant did not (Supplementary Fig. 23). Besides, the inhibition of SseK3 on the anterograde transport process was investigated by assaying the trafficking of the VSV-G protein, which is the temperature-sensitive mutant (*ts*O45) of vesicular stomatitis virus glycoprotein, in both HeLa and 293T cells. At 40.5 °C, VSV-G-GFP accumulated in the ER. When cells were shifted to the permissive temperature of 32 °C, VSV-G-GFP proteins were transported to the Golgi and further to the plasma membrane[31]. *S. flexnari* T3SS effector VirA[26] and Rab1 (S25N)[32] inhibited exocytosis and served as the positive controls. After a 4 h release into the permissive temperature, VSV-G-GFP proteins in control RFP-expressing cells shifted to the plasma membrane. However, the trafficking of VSV-G-GFP proteins in RFP-SseK3-expressing cells remained restricted in the Golgi, which showed a similar inhibition effect with Rab1A S25N expression. SseK1 and SseK3 DxD mutant did not affect the VSV-G-GFP trafficking (Fig. 5h and Supplementary Fig. 24). Consistently, VSV-G in the control cells showed increasing resistance to

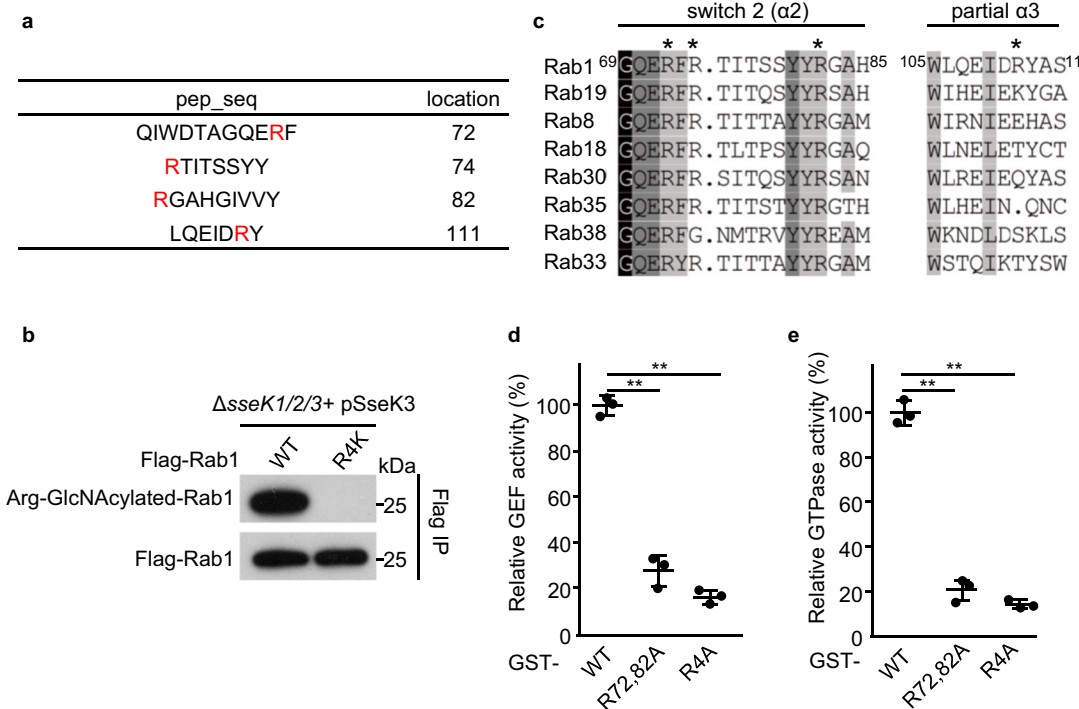

**Fig. 4 SseK3 GlcNAcylates Rab1 on Arg72, Arg74, Arg82, and Arg111 that is required for Rab1 function. a** The same samples as that in Fig. 2e and Supplementary Fig. 15 were subjected to LC-MS/MS analysis. Shown are the detected modified peptides. The red residues denote the modification sites. **b** In-vivo modification of Rab1 arginine mutants by SseK3 during *Salmonella* infection. **c** Multiple sequence alignment of the switch II region and partial α3 region of Rab small GTPases that were modified by SseK3 corresponding to Fig. 2d and Supplementary Fig. 12. **d, e** Effects of the modification site mutants on the GAP-stimulated GTPase activity of Rab1 (**d**) and the GEF activity of DrrA towards Rab1 (**e**). Recombinant proteins, GST-tagged Rab1 WT, R72/82A, and R72/74/82/111A (R4A), were subjected to GTPase activity assay supplemented with a GAP protein LepB (**d**) and GEF activity assay catalyzed by recombinant DrrA protein (**e**). The percentages of relative activity are mean ± SD from three independent experiments. **$P < 0.01$.

endoglycosidase H (Endo H) treatment along its trafficking to Golgi and post-Golgi compartments. In contrast, the sensitivity of VSV-G in either SseK3-expressing cells or VirA-expressing cells to Endo H remained sensitive after 4 h release into the permissive temperature, indicating the retention in the ER and *cis*-Golgi (Fig. 5i). Thus, SseK3 inactivates Rab1 and disrupts ER-to-Golgi trafficking through arginine GlcNAcylation.

**SseK3 inhibits cytokine secretion and is crucial for pathogenesis.** Similar to NleB, ectopic expression of SseK1 and SseK3 abolished tumor necrosis factor (TNF) treatment and TRAF2-expression-induced nuclear factor-κB (NF-κB) activation in 293T cells (Supplementary Fig. 25). Previously, other groups and us found that bacteria-delivered SseK1 and SseK3 inhibited TNF-induced cell death and NF-κB transcription due to the modification on TRADD and TNFR1, respectively[12,14,17,27]. Here we tested whether SseK effectors limit cytokine secretion during infection of macrophages. Pro-inflammatory cytokine (TNF and interleukin-6 (IL-6) release was increased in Δ*sseK1/2/3* mutant-infected supernatant compared with WT *Salmonella*-infected supernatant (Fig. 6a, b). Complementation of SseK1 and SseK3, but not SseK2- or SseK3 DxD, in the Δ*sseK1/2/3* mutant was sufficient to reduce IL-6 and TNF release (Fig. 6a, b). The amount of cytokine release is a combination of transcription–translation level and secretion level. SseK3 inactivated Rab1 and inhibited anterograde transport, which may block host cytokine secretion. We further investigated whether SseK effectors target the cytokine secretion process directly. Interferon-γ (IFN-γ) was ectopically co-expressed with SseK effectors in 293T cells. IFN-γ secretion was significantly inhibited by SseK3, but not SseK1 (Fig. 6c),

indicating that SseK3 may limit cytokine secretion on its own. As expected, the inhibitory role of SseK3 required its GlcNAc transferase activity (Fig. 6c).

Cytokines regulate host inflammatory responses, which play important roles in bacterial pathogenesis. Therefore, we investigate the roles of SseK effectors in bacterial replication and pathogenesis. The growth curve of Δ*sseK1/2/3* strain synchronizes with that of WT strain in Luria Broth (LB) medium, indicating that SseK effectors did not influence the growth characteristics of *S.* Typhimurium (Supplementary Fig. 26a). SseK effectors together were required for intracellular survival and replication within RAW264.7 macrophages but not epithelial cells (Supplementary Fig. 26b, c) and showed reduced virulence in mice compared with the WT strain at 4 days post infection. The Δ*ssaV* (an SPI-2-deficient mutant) strain was used as a control in this model (Supplementary Fig. 26d). Complementation of Δ*sseK1/2/3* with SseK1 or SseK3, but not SseK2 nor the SseK3 DxD mutant, efficiently recovered bacterial replication index in RAW264.7 macrophage cells (Fig. 6d) and restored bacterial counts in the liver and spleen of *S.* Typhimurium-inoculated C57BL/6 mice (Fig. 6e, f). Thus, the arginine GlcNAc transferase activity of SseK is essential for the inhibitory role of cytokine production and is crucial for bacterial virulence in mice.

## Discussion

In summary, we show that SseK3 localizes on the *cis*-Golgi apparatus via lipid binding and can modify a set of Rab small GTPases, especially Rab1, as the preferred host targets during *Salmonella* infection. SseK3 modifies prenylated Rab1 within the switch II region and the third α-helix, which severely disturbs the

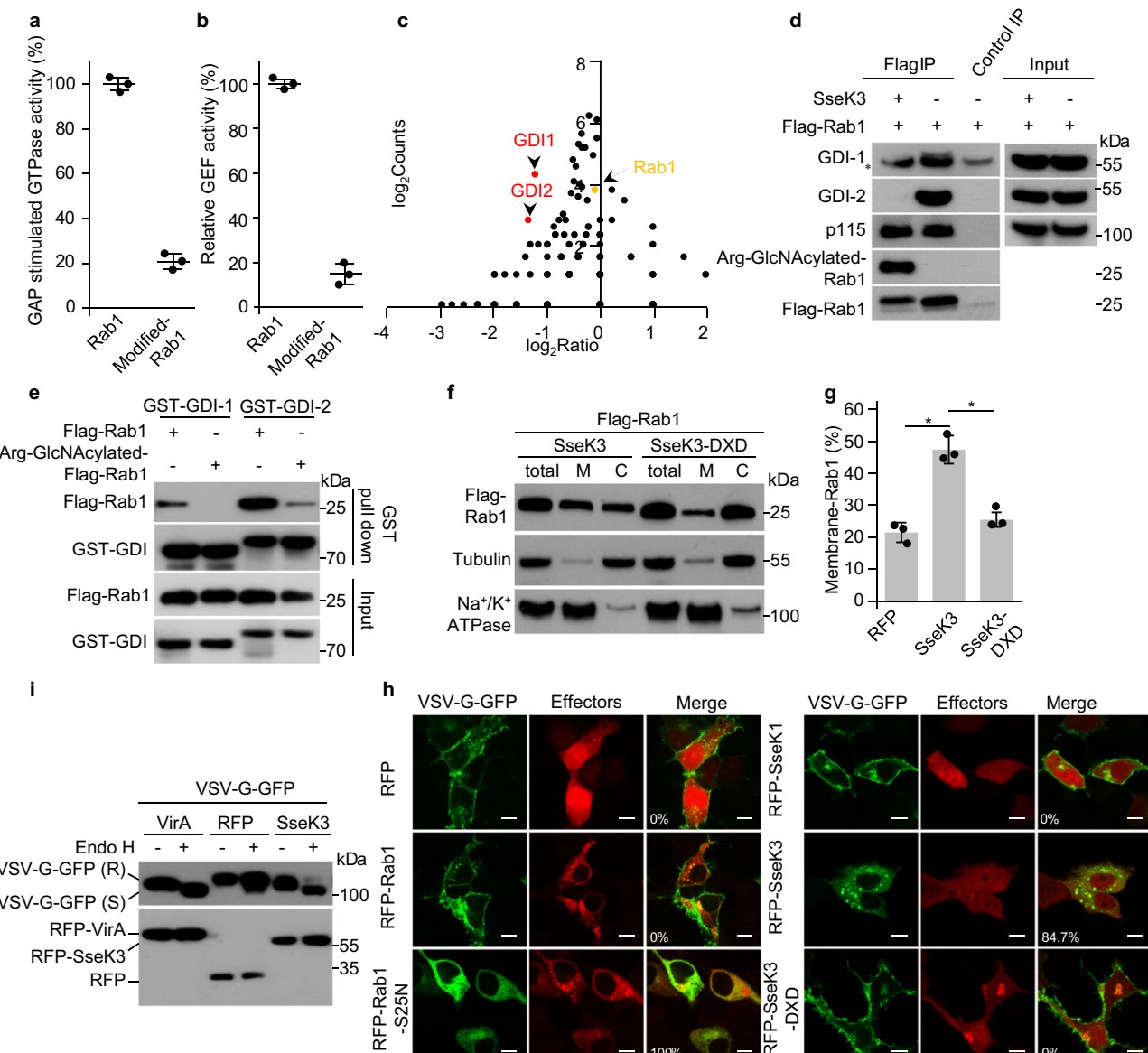

**Fig. 5 SseK3 inhibits Rab1 function and disrupts ER-to Golgi trafficking through arginine GlcNAcylation. a**, **b** Effects of arginine GlcNAcylation of Rab1 on the GTPase activity and GEF activity. Recombinant Rab1 and modified Rab1 were subjected to GTPase activity assay with the addition of GAP protein LepB (**a**) or subjected to GEF assay with the addition of GEF protein DrrA (**b**). The percentages of relative activity are mean ± SD from three independent experiments. \*\**P* < 0.01. **c–e** GlcNAcylation of Rab1 inhibits the interaction with GDIs. **c** Quantification of Rab1-binding proteins immunoprecipitated from 293T cells. Scatter plots of protein ratios as a function of their relative abundance (denoted by MS/MS spectral counts). The ratio was calculated as spectral counts in SseK3-transfected samples divided by those in mock transfection. Small ratios indicate decreased binding efficiency with Rab1 in SseK3 treated cells. Red dots correspond to GDI-1 and GDI-2, and the yellow dot corresponds to immunoprecipitated Rab1. **d** Effects of SseK3 in co-immunoprecipitation of Rab1 with endogenous GDIs. **e** Rab1 immunopurified from mock (Flag-Rab1) or SseK3 (Arg-GlcNAcylated-Flag-Rab1)-transfected 293T cells was incubated with recombinant GST-GDI1 or GST-GDI2, and subjected to GST pulldown assay. **f**, **g** Effects of arginine GlcNAcylation on the membrane cycling of Rab1. M: membrane fraction. C: cytoplasmic fraction (**f**). The ratio of the membrane-associated Rab1 was quantified using ImageJ software. The percentages of membrane-associated Rab1 are mean ± SD from three independent experiments. \*P < 0.05 (**g**). **h**, **i** SseK3 blocks VSV-G trafficking from Golgi to the plasma membrane. VSV-G-GFP-expressing HeLa cells transfected with plasmids expressing indicated RFP-tagged proteins were incubated at 40.5 °C for 16 h and then moved to 32 °C for 4 h. **h** Confocal fluorescence images of VSV-G-GFP localization are shown. Scale bars, 10 μm. Statistics of cells showing VSV-G trafficking defects are listed in the corresponding fluorescence images (at least 100 cells were counted for each experiment). **i** Cell lysates were treated with Endo H and analyzed by western blotting. R indicates the Endo H-resistant form and S indicates the Endo H-sensitive form of VSV-G.

GTPase activity, the response to GEF and GAP activity, the binding with GDIs, as well as the membrane cycle of Rab1. SseK3 inactivates Rab1 and blocks host secretory pathways, thus limiting the secretion of cytokines during *Salmonella* infection, enhancing bacterial survival in animal models. For the inhibition of cytokine production, both SseK1 and SseK3 can block NF-κB signaling by arginine GlcNAcylation on the death domain of TRADD and TNFR1, respectively. However, SseK3 but not SseK1 inactivates Rabs to hijack the secretory pathway. Previous studies show that SseK1 and SseK3 can inhibit transcription of NF-κB

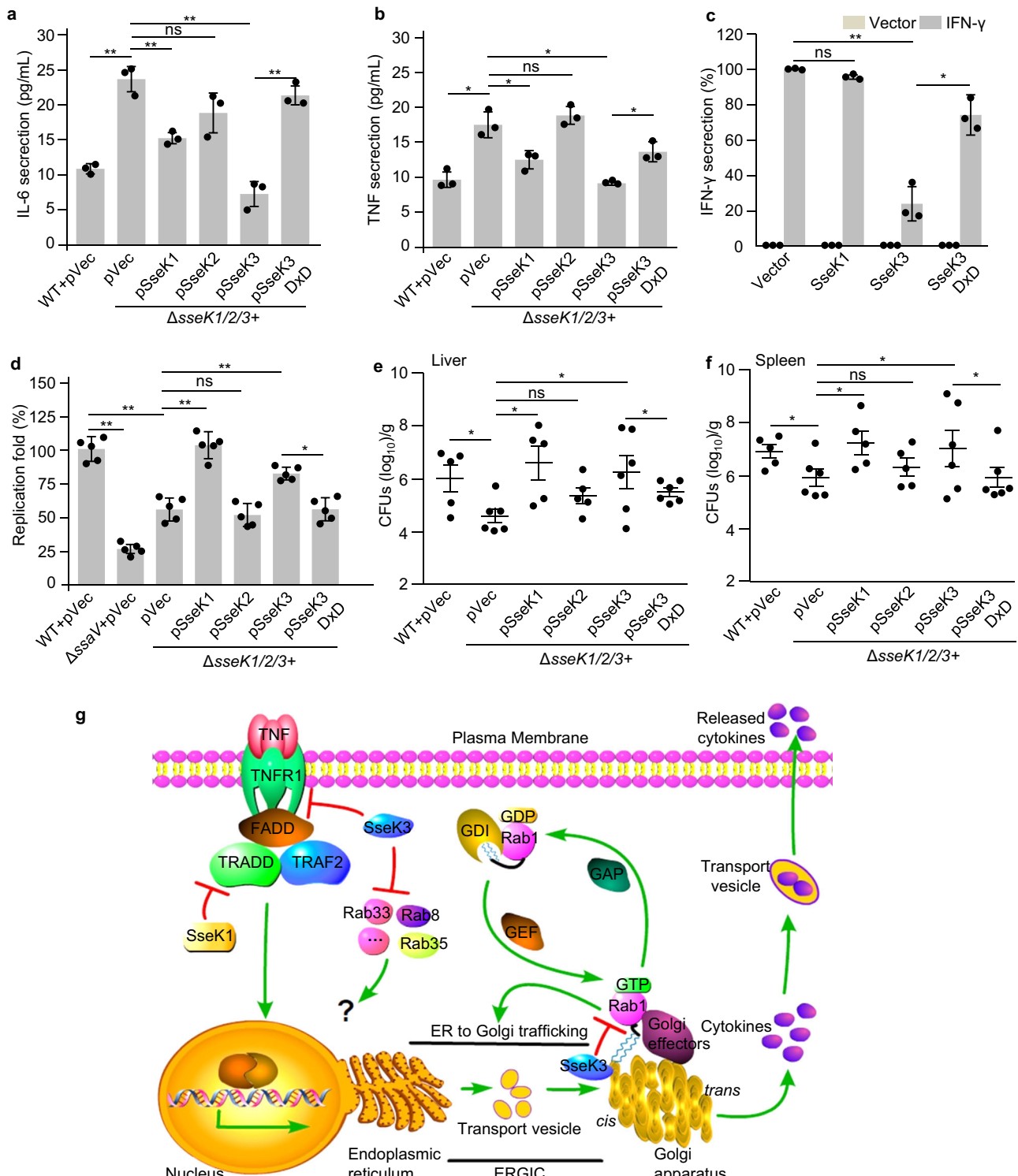

**Fig. 6 SseK3 inhibits cytokine secretion and is required for bacterial pathogenesis. a**, **b** Effects of SseK proteins on the secretion of pro-inflammatory cytokines in RAW264.7 cell during *S*. Typhimurium infection. The cytokine IL-6 (**a**) and TNF (**b**) released during 6–18 h after infection were measured in culture supernatants by ELISA and normalized to CFUs enumerated from each strain at 18 h post infection. Results shown are mean values ± SD (error bar) from three independent experiments. **c** Effects of SseK proteins or the enzymatic mutant on the secretion of IFN-γ in 293T cells. The content of IFN-γ in culture supernatant and cell lysate was quantified by ELISA and the secretion index was calculated. Results shown are mean values ± SD (error bar) from three independent experiments. **d** Effects of SseK on the replication of *S*. Typhimurium in macrophages. RAW264.7 were infected with the indicated *S*. Typhimurium at a multiplicity of infection of 10. Fold replication was determined by comparing bacterial counts at 2 and 24 h post infection. Results shown are mean values ± SD (error bar) from five independent experiments. **e**, **f** Effects of SseK proteins on the bacterial virulence in vivo. Six-week-old C57BL/6 mice were orally infected with the indicated *S*. Typhimurium strains and killed at 4 days post infection. Bacterial counts in the liver (**e**) and spleen (**f**) were calculated as colony-forming units (CFUs) per gram of tissue. A minimum of five mice was used for each group. Results shown are mean values ± SEM (error bar). n.s., not significant. *$P < 0.05$, **$P < 0.01$. **g** The schematic diagram of this work.

signaling reporter genes during *Salmonella* infection[17]. Therefore, SseK3 can limit cytokine production both at the level of transcription–translation and secretion (Fig. 6g).

The SseK3 protein colocalizes with the Golgi structure mainly due to PtdIns(4,5)P$_2$ binding. However, we do not exclude other possibilities contributing to the SseK3 Golgi localization. The SseK3 DxD mutant needed an additional 6 h to localize on the Golgi structure with the same extent to WT SseK3 (Supplementary Fig. 1). According to our research, SseK3 modified Rab small GTPases, disturbed the membrane-trafficking cycle, and trapped the VSV-G protein in the Golgi apparatus. Hence, the enzymatic activity of SseK3 may somewhat contribute to the Golgi localization of itself.

SseK3 exhibits a markedly higher enzymatic activity towards Rab1 in 293T cells than that in *E. coli*. 293T cell-purified but not *E. coli*-purified-Rab1 can be modified by SseK3 in reconstitution system. Prenylation signal (CC-SS or ΔCC) mutants of Rabs cannot be modified by SseK3 neither in ectopic transfection nor in *Salmonella* infection. However, we do not know the effect of prenylation is direct or indirect. It is possible that other eukaryotic cell-dependent modifications of Rab1 are required beyond prenylation.

Several *Salmonella* effectors have been reported to modulate the activity of host GTPases by mimicking host regulators or direct cleavage, such as SopE/SopE2 (GEF activity)[33], SptP (GAP activity)[34], SopD2 (GAP activity)[35], and GtgE (protease activity)[36]. However, covalent modification, another important strategy adopted by bacterial proteins to regulate host GTPases, is much less understood in *Salmonella*[37]. To our knowledge, SseK3 is the first identified *Salmonella* SPI-2 effector functioning in the alteration of host GTPases by covalent modification. Similar to SseK3, bacterial effectors from *Legionella* disturb Rab1 function by covalent modification at residues within switch II region. AnkX phosphorylates Rab1 at Ser76 and affects its activity in GTP loading stimulated by SidM (GEF) and hydrolysis induced by LepB (GAP)[38]. AMPylation on Tyr77 by DrrA disrupts the ability of Rab1 to mediate vesicle transport within the secretory pathway in eukaryotic cells[39,40]. SetA glucosylates Rab1 at Thr75, thus attenuating its GTPase activity and inhibiting its interaction with the GDI-1[41]. Therefore, these findings stimulate our thinking of a new strategy adopted by *Salmonella* to target the host trafficking system. These *Salmonella* T3SS effectors can disturb the activity of several Rab GTPases, such as Rab1, Rab8, Rab33, Rab29, Rab32, and Rab38. Thus, a possible future direction is to dissect the potential interplays among these effectors and how each is temporally and spatially regulated to ensure successful infection.

## Methods

**Plasmids, antibodies, and reagents**. SseK1, SseK2, and SseK3 DNA were amplified from the genomic DNA of *S.* Typhimurium strain SL1344 and inserted into the pCS2-EGFP, pCS2-HA, and pCS2-Flag vectors for transient expression in mammalian cells, or the pET28a and pGEX-6p-2 vectors for recombinant expression in *E. coli*. For complementation in *S.* Typhimurium Δ*sseK1/2/3* strain, SseK1, SseK2, and SseK3, together with their upstream promoter regions, were amplified from *S.* Typhimurium SL1344 genomic DNA and inserted into pET28a. cDNAs for GM130 and Golgin-84 were amplified from human ultimate open reading frame (ORF) clones (Invitrogen). cDNAs for Sec22B, GDI-1, GDI-2, IFN-γ, PIP4P1, Fig4, and MTM1 were amplified from a HeLa cDNA library. For mammalian expression, cDNAs were cloned into pCS2-EGFP and pCS2-Flag vectors. Truncation, deletion, and point-mutation mutants were constructed by the standard PCR cloning strategy. All plasmids were verified by sequencing.

The anti-Arg-GlcNAc antibody (ab195033, Abcam) was described as previously[42]. Antibodies for α-tubulin (T5168), β-actin (A2066), and Flag M2 (F7425) were purchased from Sigma-Aldrich. Enhanced GFP (EGFP; sc8334) and red fluorescent protein (RFP) antibodies (PM005) were obtained from Santa Cruz Biotechnology and MBL, respectively. Antibodies for glutathione *S*-transferase (GST) (26H1, #2624S), His-tag (27E8, #9991S), and Rab1 (D3X9S, #13075) were from Cell Signaling Technology. Anti-GM130 (5239872), anti-p230 (#611280), and

anti-HA Epitope Tag (901501) antibodies were from BD and Biolegend. Antibodies for Dnak (8E2/2, ab69617), Mannosidase II (ab12277), and LAMPI (ab24170) were from Abcam. The anti-Rab5 (66339-1-Ig) antibody was purchased from Proteintech. All the cytokines enzyme-linked immunosorbent assay (ELISA) kits were purchased from Proteintech. Nocodazole was from Selleckchem. Cell culture products were from Invitrogen. Rapamycin and other chemicals were obtained from Sigma-Aldrich, unless stated otherwise.

**Cell culture, transfection, and stable cell-line construction**. 293T, HeLa, MEF, and RAW264.7 cells were obtained from the American Type Culture Collection and were maintained in Dulbecco's modified Eagle's medium (DMEM) (HyClone) supplemented with 10% fetal bovine serum (FBS) (Gibco), 2 mM L-glutamine, 100 U mL$^{-1}$ penicillin, and 100 μg mL$^{-1}$ streptomycin. Cells were cultivated in a humidified atmosphere containing 5% CO$_2$ at 37 °C. Transient transfection was performed using Vigofect (Vigorus) or Jetprime (Polyplus) reagents following the manufacturer's instructions.

To generate Flag-Rab1 stable expression cells, pWPI-Rab1 WT, pWPI-Rab1 R4K, or pWPI-Rab1 ΔCC plasmid was co-transfected with packaging plasmid psAX2 and envelope plasmid pMD2.G into 293T cells. The transfection cocktail was removed after 6 h and replaced by fresh medium. After 72 h, viral containing supernatant was collected, 0.45 μM filtered, and stored at 4 °C before infection. Polyclonal populations of Rab1-overexpressed 293T cells were generated by infection with lentiviral particles. After 3 days post infection, the green fluorescent cells were sorted by fluorescence-activated cell sorting and cultured in DMEM medium supplemented with 10% FBS and 1% v/v penicillin/streptomycin.

**Bacterial strains and cell culture infection**. The *S.* Typhimurium strains (WT SL1344 and its mutant derivatives) are used in this study. *S.* Typhimurium Δ*sipD*, Δ*ssaV*, and Δ*sseK1/2/3* were generated by a standard homologous recombination method using the suicide plasmid pCVD442. pME6032 or pET28a vector-based complementation plasmids were introduced into *S.* Typhimurium by electroporation (2.5 kV, 200 Ω, 25 μF, and 5 ms). Bacteria were cultured in LB broth at 37 °C with shaking. When necessary, cultures were supplemented with antibiotics with the following final concentrations: streptomycin, 100 μg mL$^{-1}$; ampicillin, 100 μg mL$^{-1}$; and kanamycin, 50 μg mL$^{-1}$.

The procedure for bacterial infection of mammalian cells was as previously described[43]. Briefly, WT and mutant *Salmonella* were cultured overnight (~16 h) at 37 °C with shaking and then subcultured (1 : 33) in LB without antibiotics for 3 h. Bacterial inoculates were diluted in serum-free and antibiotics-free DMEM, and added to cells at a multiplicity of infection (MOI) of 100 for 30 min at 37 °C. Twenty-four-well plates were centrifuged at 700 × *g* for 5 min at room temperature to promote and synchronize infection. Extracellular bacteria were removed by extensive washing with phosphate-buffered saline (PBS) and culture media was replaced with media containing 100 μg mL$^{-1}$ gentamicin. Cells were incubated at 37 °C, 5% CO$_2$ for a further 1.5 h, and the culture media was replaced with media containing 20 μg mL$^{-1}$ gentamicin. Infected cells were incubated to the indicated time at 37 °C in a 5% CO$_2$ incubator and subjected to further immunoprecipitation or immunofluorescence.

To measure the *S.* Typhimurium replication fold, RAW264.7 cells (with about 90% confluency) plated in 24-well plates were infected with indicated *Salmonella* strains at an MOI of 10. Infection was facilitated by centrifugation at 700 × *g* for 5 min at room temperature. After 30 min incubation at 37 °C, cells were washed three times with PBS to remove extracellular bacteria and incubated with fresh DMEM containing 100 μg mL$^{-1}$ gentamicin. At 2 h post infection, the gentamicin concentration was reduced to 20 μg mL$^{-1}$. At 2 and 24 h post infection, cells were lysed in cold PBS containing 1% Triton X-100 and colony-forming units were determined by serial dilution plating on agar plates containing 100 μg mL$^{-1}$ streptomycin and 50 μg mL$^{-1}$ kanamycin. The replication fold was determined by dividing the number of intracellular bacteria at 24 h by the number at 2 h.

**PIP Strip-binding assay**. The lipid-binding assay was performed with PIP Strip (P-6001) (Echelon Biosciences) according to the manufacturer's instruction. The concentration of purified GST, GST-SseK3, and GST-SseK3 (K87A/R89A/K234A) protein is 1 μg/ml in blocking buffer (PBS with 0.2% Tween 20 and 3% bovine serum albumin (BSA)). All the steps were performed at room temperature.

**Inducible recruitment of phospholipid phosphatases**. Plasmids pTGN38-FRB-HA and pmRFP-FKBP12-Sac1 were obtained from Addgene. We replaced the *sac1* gene in pmRFP-FKBP12-Sac1 with other phosphatase genes to construct four plasmids: pmRFP-FKBP12-PIP4P1, pmRFPFKBP12-lipin1, pmRFPFKBP12-MTM1, and pmRFP-FKBP12-Fig4. In brief, HeLa cells in 24-well plates were transiently transfected with pCS2-EGFP-SseK3 (0.2 μg), pTGN38-FRB-HA (0.15 μg), and pmRFP-FKBP12-phosphatase (0.15 μg) together using JetPrime (Polyplus) reagents following the manufacturer's instructions. The cells were treated with rapamycin (2 μM) for 120 min before immunostaining with anti-GM130 antibody and Alexa Fluor 647-conjugated secondary antibody, and subsequently imaged with a confocal microscope (Spinning Disc, Leica).

**Mice infection and *S.* Typhimurium virulence assays**. Five- to 6-week-old WT C57BL/6 mice were purchased from Liaoning Changsheng Biotechnology Co. and were maintained in the specific pathogen-free facility at Huazhong Agricultural University. All animal experiments were carried out in accordance with the Ministry of Health national guidelines for housing and care of laboratory animals, and were performed in accordance with institutional regulations after review and approval by the Institutional Animal Care and Use Committee at Huazhong Agricultural University. Mice were randomized into each experimental group with no blinding. Independent experiments were performed using five to six mice per group. For infection, *S.* Typhimurium WT strain and represented derivatives were prepared by overnight shaking of bacterial culture at 37 °C in LB broth. Mice were orally inoculated using a gavage needle with 200 μL suspension of bacteria ($5 \times 10^7$ colony-forming units (CFU)) in PBS and the mice inoculated with PBS were set as a negative control. To determine the bacterial burden, mice were killed 2–4 days post infection. Tissue samples, including the respective liver and spleen contents, were removed for bacterial quantification in sterile PBS. The number of CFUs in the homogenates was determined by plating serial dilutions on agar plates. Virulence data were analyzed using a Student's $t$-test in the commercial software GraphPad Prism 5.0 and $P$-values ≤ 0.05 were considered significant.

**Immunoprecipitation**. For co-immunoprecipitation, 293T cells at a confluency of 60–70% in 6-well plates were transfected with a total of 5 μg plasmids. Twenty-four hours after transfection, cells were washed once in PBS and lysed in buffer A containing 25 mM Tris-HCl pH 7.5, 150 mM NaCl, 10% glycerol, and 1% Triton X-100, supplemented with a protease inhibitor mixture (Roche Molecular Biochemicals). Precleared lysates were subjected to anti-Flag M2 immunoprecipitation following the manufacturer's instructions. The beads were washed four times with lysis buffer and the immunoprecipitates were eluted by SDS sample buffer followed by standard immunoblotting analysis. All the immunoprecipitation assays were performed more than three times, and representative results are shown[12].

To enrich the Arginine-GlcNAcylated proteins from transfected or infected cell lysate, cells were washed three times in ice-cold PBS and lysed in buffer A containing 25 mM Tris-HCl pH 7.5, 150 mM NaCl, 10% glycerol, and 1% Triton X-100, supplemented with a protease inhibitor mixture (Roche Molecular Biochemicals). Precleared lysates were subjected to anti-N-GlcNAc immunoprecipitation. The beads were washed four times with lysis buffer and the immunoprecipitates were eluted by SDS sample buffer.

To purify Flag-Rab1 for total molecular MS, Rab1 stably expressed 293T cells were collected in lysis buffer B containing 50 mM Tris-HCl pH 7.5, 150 mM NaCl, 20 mM $n$-octyl-β-D-glucopyranoside (INALCO), and 5% glycerol, supplemented with an EDTA-free protease inhibitor mixture (Roche Molecular Biochemicals). Cells were lysed by ultrasonication and precleared lysates were incubated with Flag M2 beads. After a 4 h incubation, the beads were washed once with buffer B and then four times with TBS buffer (50 mM Tris-HCl pH 7.5 and 150 mM NaCl), and immunoprecipitates were eluted with 600 μg mL$^{-1}$ Flag peptide (Sigma) in TBS buffer. The eluted protein was verified by Coomassie brilliant blue staining or silver staining on an SDS-polyacrylamide gel electrophoresis (PAGE) before MS analysis.

**Expression and purification of recombinant proteins**. Protein expression was induced in *E. coli* BL21 (DE3) strain (Novagen) at 22 °C for 15 h with 0.4 mM isopropyl-β-D-thiogalactopyranoside after absorbance at 600 nm ($A_{600 \text{ nm}}$) reached 0.8–1.0. Affinity purification of GST-SseK3, GST-GDI-1, and GST-GDI-2 proteins were performed using glutathione sepharose (GE Healthcare), and purification of 6× His-SUMO-Rab1 and 6× His-Rab1 was conducted using Ni-NTA agarose (Qiagen), following the manufacture's instructions. His-SUMO (6 ×) was removed by ULP1 protease digestion after affinity-chromatography purification. Proteins were further purified by HiTrap Q HP ion-exchange chromatography and size-exclusion chromatography (GE Healthcare), and concentrated in a buffer containing 20 mM HEPES pH 7.5, 150 mM NaCl, and 5% glycerol. The protein concentration was determined by Coomassie blue staining of SDS-PAGE gels using BSA standards.

**Detection of cytokine production**. The concentration of IL-6, TNF, and IFN-γ in culture supernatants or cell lysates was measured with a mouse IL-6 ELISA kit (Proteintech), a mouse TNF ELISA kit (Proteintech), or a human IFN-γ ELISA kit (Proteintech), following the manufacturer's instructions.

We adopted the reported methods for the detection of cytokine production during *Salmonella* infection[44]. Briefly, RAW264.7 cells were seeded in 12-well plates and infected as described above. At 18 h post infection, the cultural supernatants were assayed for IL-6 and TNF levels using ELISA kits. Data were shown as the average cytokine concentration (pg/mL) normalized to CFUs obtained for each strain at 18 h post infection. Assays were done in three separate experiments and statistical analyses were performed using a two-tailed Student's $t$-test.

To test the effects of SseK proteins on the secretion of IFN-γ, 293T cells were transfected with 1 μg pCS2-GFP-effector or pCS2-GFP control plasmids with or without 500 ng pCS2-1 × Flag- IFN-γ. At 24 h after transfection, the contents of IFN-γ in the supernatant and cell lysate were quantified by ELISA and the secretion

index was calculated. In addition, the secretion index changes of IFN-γ between GFP control and effectors were calculated.

**hGH-trafficking and VSV-G transport assays**. The hGH-trafficking assays were performed as described previously[45]. 293T cells plated in 12-well dishes were transfected with 1.2 μg of 4 × FKBP-hGH (Ariad Pharmaceutical) and 1.2 μg of pCS2-GFP vector expressing the target proteins. Eighteen hours post transfection, the culture media was replaced with media containing 0.5 μM D/D Solubilizer (Clontech) or ethanol as a control, and cells were incubated for another 4 h. The concentration of hGH in culture supernatants or cell lysates was measured using a hGH ELISA kit (Roche Life Science) according to the manufacturer's instructions.

To monitor VSV-G trafficking from the ER to the plasma membrane, 293T and HeLa cells cultured on coverslips in 24-well dishes were transfected with a plasmid expressing VSV-G-GFP (*ts*O45) together with a plasmid expressing target proteins. Transfected cells were incubated at 40.5 °C for 16 h and then transferred to 32 °C to initiate VSV-G-GFP release from the ER. Four hours after release, the cells were fixed with 4% paraformaldehyde (PFA) and fluorescence images were captured. To examine the sensitivity of VSV-G to Endo H digestion, the released 293T cells were collected in 1× glycoprotein-denaturing buffer and the lysates were treated with 100 U of Endo H (New England Biolabs) at 37 °C for 2 h. Reactions were terminated by adding SDS loading buffer. Boiled samples were then loaded onto a 10% SDS-PAGE gel and analyzed by anti-GFP immunoblotting[46].

**GDI-pulldown assay**. GST-GDI-1 and GST-GDI-2 proteins were expressed in the *E. coli* BL21(DE3) strain and purified using glutathione resin as described above. pCS2-RFP-SseK3 or pCS2-RFP-SseK3 (DxD) plasmid was co-transfected with pCS2-Flag-Rab1 into 293T cells. After 18 h, cells were lysed in the GST-pulldown buffer containing 25 mM Tris-HCl pH 7.5, 150 mM NaCl, 25 mM β-glycerophosphate, 1 mM sodium orthovanadate, 10% glycerol, 0.5 mM dithiothreitol (DTT), 1 mM phenylmethylsulphonyl fluoride, and 1% Triton X-100, supplemented with the protease inhibitor mixture. Total cell lysates were incubated with GST-GDI-immobilized glutathione resins for 1 h at 4 °C. The beads were washed once with PBS plus 1% Triton X-100 and twice with PBS plus 0.5% Triton X-100. Bead-bound proteins were analyzed by immunoblotting using indicated antibodies.

**In-vitro GTPase assay and GEF assay**. The GTPase assay and GEF assay were performed using Promega's GTPase-Glo™ Assay kit. For GTPase assay, 2 × GTP solution was prepared to contain 10 μM GTP and 1 mM DTT with or without 4 μg LepB in GTPase/GAP Buffer. Next, 400 ng purified Rab1 was diluted in GTPase/GAP Buffer. GTP solution (2×) was added and the solution incubated for 2 h to allow the GTPase reaction. The GTPase reaction systems were supplemented with 400 ng Rab, Rab1 R2A, Rab1 R4A, or modified Rab1, 4 μg LepB, and 2 μg DrrA in the GEF Buffer for GEF assay. Reactions were initiated by adding 10 μM GTP in GEF Buffer containing 1 mM DTT and incubated for 2 h at room temperature. All reactions were terminated by the addition of GTPase-Glo Reagent and then incubated for 30 min followed by the addition of Detection Solution. Ten minutes later, the luminescence was measured using a multilabel plate reader (Cytation 5, BioTek).

**Immunofluorescence labeling and confocal microscopy**. At the indicated time points post transfection or bacterial infection, cells were fixed for 10 min with 4% PFA in PBS and permeabilized for 15 min with 0.2% Triton X-100 in PBS. After blockade of nonspecific binding by incubation of cells for 30 min with 2% BSA in PBS, coverslips were incubated with the appropriate primary antibodies (mentioned above) and subsequently with fluorescein-labeled secondary antibodies (ThermoFisher). Confocal fluorescence images were acquired at the confocal microscope (Spinning Disc, Leica). All image data shown are representative of at least three randomly selected fields.

**In-vitro GlcNAcylation assay**. Rab1 proteins (200 ng) purified from *E. coli* or 293T cells were incubated with 500 ng of GST-SseK3 in 40 μL of the reaction buffer containing 1 mM UDP-GlcNAc, 2 mM MnCl$_2$, 20 mM HEPES pH 7.5, and 150 mM NaCl at 37 °C for 2 h. Reactions were terminated by boiling at 95 °C for 5 min in SDS-PAGE sample buffer. The reaction mixtures were separated by 10% SDS-PAGE and subjected to western blotting analysis.

**Fractionation of cellular membrane and cytosol**. Total membranes and cytosols of 293T cells were isolated using Plasma Membrane Protein Extraction Kit (Abcam, ab65400) according to manufacturer's instructions. Briefly, a total of 5 × 10$^6$ 293T cells were washed with cold PBS and suspended in a homogenized buffer mix in an ice-cold Dounce homogenizer (Kimble). Homogenates were centrifuged at 700 × $g$ for 10 min at 4 °C to remove the nucleus. The resulting supernatants were collected. Total membrane proteins were pelleted by centrifugation at 10,000 × $g$ for 30 min at 4 °C. The supernatant was the cytoplasmic fraction. Membrane pellets were dissolved in 0.5% Triton X-100 in PBS. The distributions of proteins in the cytosol and membrane fractions were analyzed by western blotting, and Na$^+$/K$^+$-ATPase and TubulinA served as loading controls. Band intensities of scanned blots were quantified using ImageJ.

**Mass spectrometry analyses**. For total molecular weight measurement by MS, 10 µg of *E. coli* or 293T cell-purified Rab1 protein was loaded onto a homemade C4 capillary column (MAbPacTM RP, 4 µm, 2.1 × 50 mm, Thermo Scientific, USA) packed with POROS R1 medium (Applied Biosystems). Subsequently, the proteins were eluted by an Agilent 1100 binary pump system with the following gradient: 5–100% B in 10 min at a flow rate of 0.3 mL/min (A: 0.1% formic acid in water; B: 0.1% formic acid in 80% acetonitrile/water) and sprayed into a Q Exactive Plus mass spectrometer (Thermo Scientific, USA) equipped with a Heated Electrospray Ionization (HESI-II) Probe (Thermo Scientific, USA). The instrument was acquired in MS mode under 3000 volts of spray voltage. The protein charge envelope was averaged across the corresponding protein elution peaks and deconvoluted into the uncharged form using the Protein Deconvolution program provided by the manufacturer.

To identify GlcNAcylated arginine and arginine-containing peptides, the chymotrypsin or trypsin-digested Rab1 (modified by SseK3 in bacterial or mammalian cells) was subjected to digestion and the resulting peptides were separated on an EASY-nLC 1000 system (ThermoFisher Scientific). The nano liquid chromatography gradient was as follows: 0–8% B in 3 min, 8–28% B in 42 min, 28–38% B in 5 min, 38–100% B in 10 min (solvent A: 0.1% formic acid in water, solvent B: 80% CH₃CN in 0.1% formic acid). Peptides eluted from the capillary column were applied directly onto a Q Exactive Plus mass spectrometer by electrospray (ThermoFisher Scientific) for MS and MS/MS analyses. Searches were performed against the amino acid sequence of Rab1 and were performed with cleavage specificity allowing two mis-cleavage events. Searches were performed with the variable modifications of oxidation of methionine, *N*-acetyl-hexosamine addition to arginine (Arg-GlcNAc), and acetylation of protein N termini.

To identify the Rab1-binding protein, immunoprecipitates were separated using SDS-PAGE, fixed, and visualized with silver staining as recommended by the manufacturer. An entire lane of bands was excised and subjected to in-gel trypsin digestion and MS/MS detections, as described above. The identification of proteins was accomplished using the Proteome Discoverer 2.2 program. Searches were performed against the Human or Mouse proteomes depending on the samples with carbamidomethylation of cysteine set as a fixed modification.

**Statistics and reproducibility**. All results are presented as mean ± SD containing a specified number of replicates. Data were analyzed using a Student's *t*-test. Pearson correlation coefficient (*R*) was used as a measure of the linear correlation between colocalization in fluorescence images. A difference is considered significant as the following: *$P < 0.05$ and **$P < 0.01$. In mouse infection experiments, data generated from at least five mice in each group were analyzed.

**Reporting summary**. Further information on research design is available in the Nature Research Reporting Summary linked to this article.

## Data availability

The mass spectrometry analysis raw data have been deposited in the iProx database (URL: http://www.iprox.org/page/PDV0141.html) and are available under the accession number IPX0002190000. The data will be released to public when the paper associated with this iProX ID is published. Source data of gel and blot images are included in Supplementary Data 1. Protein sequences and plasmid sequencing results have been deposited in the figshare database (https://figshare.com/s/78544f0f1453822fe202). All other relevant data supporting the findings of this study are available in Supplementary Data 2.

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

## Acknowledgements

We thank members of the Li laboratory at Huazhong Agricultural University (HZAU) and the Institute of Infection and Immunity in Taihe hospital for helpful discussions and technical assistance. This work was supported by the National Key Research and Development Programs of China 2018YFA0508000, Foundation for Innovative Research Team of Hubei Provincial Department of Education T201713, Fundamental Research Funds for the Central Universities 2662017PY011, 2662018PY028, 2662019YJ014, and 2662018JC001, and Huazhong Agricultural University Scientific & Technological Self-Innovation Foundation 2017RC003 to S.L.

## Author contributions

S.L. and K.M. conceived the study. K.M., X.Z., and S.H. designed and performed the functional experiments. T.P. and K.M. determined the localization of SseK3 on *cis*-Golgi via lipid binding. T.P. and X.Z. performed the in-vitro GlcNAcylation reactions. J.Y. determined the localization of SseK3 with the SunTag system. X.P., J.X., Z.W., J.F., and X.L. provided technical assistance of mass spectrometry. J.L. constructed the deletion mutant strain of *Salmonella*. K.M. and S.L. analyzed the data and wrote the manuscript. All authors discussed the results and commented on the manuscript.

## Competing interests

The authors declare no competing interests.
