## [Peer Review File · Communications Biology]

Reviewers' comments:

Reviewer #1 (Remarks to the Author):

Communications Biology COMMSBIO-19-1941-T

Title: Arginine GlcNAcylation of Rab small GTPases by the pathogen *Salmonella typhimurium*

The manuscript entitled "Arginine GlcNAcylation of Rab small GTPases by the pathogen *Salmonella typhimurium*" by Dr. Li and colleagues reported the role of Ssek3, a salmonella-derived arginine GlcNAc (R-GlcNAc) transferase, in disrupting Golgi function via R-GlcNAcylation of Rab proteins. Previously, the same author identified that NleB1, an enteropathogenic *E. coli* (EPEC)-derived R-GlcNAc transferase, disrupts TNF signaling via R-GlcNAcylation of TRADD protein. In this study, the author found a Golgi compartment location of Ssek3 after salmonella infection, which leads to a disruption in the Golgi structure. Through mass spec analysis of R-GlcNAcylated proteome after salmonella infection, the authors identified a series of Rab proteins as major targets of Ssek3-mediated R-GlcNAcylation. Functionally, R-GlcNAcylation of Rab1 results in the loss of its GTPase activity and a defect in ER-to-Golgi protein transport. Furthermore, the authors showed that Ssek3 is critical for inhibition of cytokine release from salmonella-infected macrophages, as well as the in vivo pathogenicity of salmonella. Overall, the experimental strategies, including various imaging and biochemical assays, were well designed and the data were compelling. However, the role of salmonella Ssek3 in macrophage cytokine production is still confusing. Some specific comments are listed below.

- In Figure 4 b, d and e, the authors showed the loss of R-GlcNAcylation of Rab1 when four R residues were mutated to lysine and the loss of GTPase activity/GEF activity when four R residues were mutated to alanine. What is the reason to perform assays using different mutations? The reviewer is surprised that both GTPase activity and GEF activity of Rab1 were defective when R-GlcNAcylation was removed, which indicates an unspecific inhibitory effect of Ssek3-mediated R-GlcNAcylation on Rab1 function. This phenomenon is different from other effector salmonella effectors (references 31-33). Could the authors add some discussion on this issue?
- In Figure S5, the authors showed that Ssek3 over expression led to a disruption of Golgi structure. The experimental setup was quite different from Figure 1. It would be more ideal to compare the effect of WT, delta ssek1/2/3, and delta ssek1/2/3 plus Ssek3 on Golgi structure, as performed in Figure 1.
- In Figure 5, the authors performed a series of imaging and biochemical assays to conclude that Ssek3 inhibits Rab1 function (binding GDI-1/2, member-cytosol recycling). One most significant assay should be testing Rab1 function and protein ER-to-Golgi transport using R-GlcNAcylation-deficient Rab1 mutants (R4K or R4A) since they are available from the Figure 4. These assays will be complement to the assays using enzyme dead mutant of Ssek3 (Fig 5h, right panel).
- The most significant weakness of the manuscript is Figure 6. The reviewer is still confused by the individual roles, if there is any, of Ssek3 in regulates bacterial-induced cytokine production versus release.
 - First, the ELISA data of IL-6 and TNF are less than 30 pg/ml in macrophage cell culture after salmonella stimulation for 6-18 h. It is hard to understand why the numbers are so low. What did it mean "...normalized to CFUs enumerated from each strain at 18 h post-infection"?
 - Second, how does Ssek1 or 3 inhibits cytokine production in macrophages upon salmonella challenge? If it is related to signaling transduction, which target(s)? I do not think TNF signaling has something to do with salmonella-elicited cytokine production. It should be TLR signaling, for most of parts.
 - Third, what is the transcript levels of IL-6 and TNF in Fig. 6a and b? The results may help to distinguish the scenarios of cytokine production versus cytokine release.

- Fourth, IFN-g is a T cell cytokine, I do not feel it will add any significance for the physiological role of Ssek3 in cytokine secretion to artificially use a salmonella-T cell cytokine system.
- Finally, in Fig 6d, it is not a usual way to show intracellular growth of salmonella like replication fold. A growth curve will be helpful. Meanwhile, I am still not sure if any difference in intracellular bacterial growth is due to the cytokine difference. It is a unapproved assumption.

Reviewer #2 (Remarks to the Author):

Arginine GlcNacylation of Rab small GTPases by the pathogen Salmonella Typhimurium

In this work, Kun Meng et al. present data to support the arginine GlcNacylation of Rab GTPases by the effector SseK3 and they dissect the role of this PTM in trafficking during Salmonella infection. Although the data presented adds to our understanding of this understudied PTM, several points are not solidly supported by experimental observations and the quality of the manuscript needs to be improved significantly. It is poorly written and the authors should engage with native English speakers to improve readability.

The authors infect transformed epithelial cellular models containing with sun-tag modified SseK proteins to imitate previously published infectious models on macrophages. However, the provided data was not obtained from a single cell line, HeLa and 293T cells were used to complete different experimental and mixing data from two immortalized cells lines could lead to the wrong interpretations. If two cell lines are to be used, proper controls to assure that the observed phenotypes are conserved in both cellular systems are required to validate the presented data. It is also not clear why these non-physiological systems were used if primary cells/more appropriate cell lines are available.

SseK3 localises on cis-Golgi via lipid binding.

In the Supplementary figure 1b the only control used is an empty vector. However, the inactive SseK1,3 controls are missing. This is important as the used sun tag plus the 24 GFP molecules weigh more than 1000KDa and this could lead to migration artefacts.

To explain the observed location of SseK3 in the Golgi apparatus the authors identified an electronegative patch constituted by the residues K87/R89/K234, the latter two conserved in the NleB1 orthologue which doesn't migrate to the Golgi. Figure 1e shows a reduced but not abolished binding to the PtdIns(4,5)P2 for the K87A/R89A/K234A mutant. However, presented data shows strong binding to PI3P, PI5P and probably to PI(3,5)P2 regardless of the presence of the K87A/R89A/K234A mutations. This, together with the ability of sseK3 to, still, being recruited to the Golgi under rapamycin treatment suggest other interactions outside the proposed electronegative patch (or other protein factors) and supports a non-disclosed recruitment path that might work independently of the PtdIns(4,5)P2 presence. Suggesting that, in opposition to authors claim, PtdIns(4,5)P2 is not essential for SseK3 recruitment to the Golgi. This requires further experimental clarification.

In supplementary figure 3a, the electrostatic potential scale and the method for calculating the values are missing.

Prenylation of Rab1 is crucial for Arginine GlcNAcylation by SseK3.

The authors state that Rab1 prenylation is important for the arginine modification but the presented experimental setup can only prove that silencing the Rab1 prenylation signal (CC) abolishes trafficking to the Golgi. The authors imply a direct effect but this might as well in indirect (i.e. involving other prenylated protein partners). Furthermore, Rab1 could potentially be modified by additional modifications (PTM) which makes it a target for SseK3. To clarify this issue intact

mass spec to determine the presence/absence of additional PTM in the Δ CC and the CC-SS vs the WT is required.

Figure 3a is not clear. The second lane shows GlcNAcylated Rab1 but no signal from Rab1, same for the third lane. Several lanes are shown increased signal for GlcNAcylated Rab1 but no explanation is given for these effects.

In Figure 3d, would be useful to indicate which sample was recombinantly produced instead of expecting the reader to guess it by the Rab1 molecular size shift.

SseK3 modifies Rab1 on Arg72, Arg74, Arg82 and Arg111.

The authors only use mass spectrometry data and site point SseK3 mutants to search and validate the putative Rab1 modification sites. Arg72, Arg74, Arg82 and Arg111 are proposed based on their experimental data. However, the structural architecture of Rab1 makes it highly improbable that these sites are modified. Further data, like in vitro GlcNAc transfer activity assay onto Rab1 based peptides is required to validate this observation, especially since the quadruple R4K mutant referred in Supplemental Figure 4b or the mutants analysed in Supplemental Figures 4d and 4e does not provide clarification which is/are the real Rab1 glycosylation sites.

Furthermore, the authors get confused and referred to main text figures while the data is instead shown in supplemental files. (lines 257 to 262). This and many other errors in the text makes it very hard to understand their findings.

SseK3 inactivates Rab1 through Arginine GlcNAcylation and disrupts ER-to Golgi trafficking.

Figure 5a and 5b. Bar representations are not appropriate to assess the quality of the data, scatter plots are required instead.

Text and figures often mention the vague term "mammalian cells", which is not helpful for the reader. Would be better to indicate which specific cell lines were used instead.

SseK3 is crucial for pathogenesis by inhibiting cytokine production and secretion.

The RAW264.7 cell line is used to check for the replication of WT *S. Typhimurium* and Δ sseK1/2/3 mutant. However, RAW264.7 is a murine cell line and human counterpart/primary cells would be useful as a validation control.

Dear reviewers,

We are genuinely grateful to your critical comments and thoughtful suggestions on our manuscript entitled “**Arginine GlcNAcylation of Rab small GTPases by the pathogen *Salmonella Typhimurium***” (COMMSBIO-19-1941A). This will be very helpful for us to improve the quality of our manuscript.

In this response letter, we try our best to answer these questions with on-hand experimental data, which are not shown in the first version manuscript. Only three questions, numbered as Q2, Q8, and Q9 in the response letter, cannot be answered by furthermore experiments immediately. The reason why we cannot do any bench work now is very special.

As you may know, many people in China, especially in the Hubei province, are suffering from the new coronavirus and COVID-19. All the experiments in this study are finished in Hubei province, the cities Wuhan and Shiyan. All the authors of this manuscript are quarantined in their hometown and cannot come back to the lab for at least two months. Furthermore, the Bio-Safety Level-2 lab, where we did all the bacterial infection experiments, are being used to manipulate the nucleic acid of the new coronavirus. Therefore, to avoid the risk of cross-infection, unrelated researchers and all other experiments are forbidden in those BSL-2 rooms now. Unfortunately, there is not a clear timeline for these changes. We are grateful that people all over the world are fighting for this epidemic together. However, for this manuscript, there are fierce competitions, and we know a similar story about SseK3 protein is ongoing.

In this response letter, we answer most of the questions with experimental data and would like to ask if there is any possibility of accepting the response as it is now. Below you will find our point-by-point responses to your comments/ questions. The editor’s questions are in blue and bold, following by the reviewer’s corresponding questions that are in blue. Our response, additional figures, and figure legend, to every question/comment, are immediately below your questions.

Q1. Do the experiment of S5 as Fig 1 to enable good comparison (by comparing WT and different mutation backgrounds on Golgi structure) (Reviewer 1).

Corresponding to the question: In Figure S5, the authors showed that Ssek3 over expression led to a disruption of Golgi structure. The experimental setup was quite different from Figure 1. It would be more ideal to compare the effect of WT, delta ssek1/2/3, and delta ssek1/2/3 plus Ssek3 on Golgi structure, as performed in Figure 1.

We agree with the reviewer that it is necessary to examine if the Golgi structure was disrupted by *Salmonella*-infection-delivered-SseK3.

According to Fig1a and Supplementary Fig 1, 8-14 h after *Salmonella* infection on HeLa cells, the Golgi structure was intact, and *Salmonella*-delivered-SseK3 co-localized with the Golgi structure (shown as scFv-GCN4-GFP and GM-130 staining in Supplementary Fig 1). To address the SseK3 involvement in disrupting the Golgi structure we performed three tests. First, we extended infection and observation time, and

found that at 24 h post $\Delta sseK1/2/3$ plus SseK3 strain infection, the Golgi structures in over 60% of infected cells were disrupted (shown as GM-130, ManII, and Rab1 staining). Although we do not know the effect of WT *Salmonella* infection immediately, we can observe the Golgi disruption upon *Salmonella* infection. This disruption was dependent on SseK3 delivery by comparing the effect of $\Delta sseK1/2/3$ and $\Delta sseK1/2/3$ plus SseK3. (Fig R1a,b). Secondly, we dissected the SseK3 protein level in both ectopic expression and the *Salmonella* delivering system. We observed the phenomenon existed in both systems, that Golgi structure stayed intact in cells expressing SseK3 at a relatively low level and was disturbed in cells expressing SseK3 at a relatively high level (Fig R2). Thirdly, we detected the arginine GlcNAcylation level upon *Salmonella* infection. This SseK3 catalyzed PTM accumulated over time (Fig R3). So *Salmonella* infection can trigger Golgi disruption in a SseK3 dependent manner. Meanwhile, we speculate that the Golgi structure disruption is dependent on the SseK3 protein level and Arginine GlcNAcylation accumulation. These figures will be further arranged and added to supplementary data.

Fig R1 SseK3 disturbed Golgi structure 24 h post *Salmonella* infection. a,b, Effects of SseK3 on endogenous Golgi protein immunostaining during *Salmonella* infection. **a**, HeLa cells were infected with *S. Typhimurium* $\Delta sseK1/2/3$ complemented with empty vector or SseK3-expressing plasmid as indicated for 24 h. Shown are immunofluorescence staining using anti-GM130, anti-ManII or anti-Rab1A antibodies, and DAPI indicated the host and bacterial DNA. **b**, Statistics of cells are listed according

to **a**, and the percentages of Golgi disruption are mean \pm SD from three determinations. At least 100 cells were counted for each experiment.

Fig R2 Effects of SseK3 on the morphology of the Golgi structure. **a**, HeLa cells were transfected with a plasmid expressing GFP-SseK3 (left), or infected with *S. Typhimurium* Δ *sseK1/2/3* complemented with SseK3-Flag-expressing plasmid for 24 h (right). Shown are immunofluorescence staining using anti-GM130 (left), anti-Rab1, or anti-Flag antibodies (right) in cells expressing different levels of SseK3 protein. **b**, Statistics of cells are listed according to **a**, and the percentages of Golgi disruption are mean \pm SD from three determinations. At least 100 cells were counted for each experiment.

Fig R3 Arginine GlcNAcylation Pattern catalyzed by SseK3 during *Salmonella* infection. 293T cells were infected with *S. Typhimurium* Δ *sseK1/2/3* complemented with an empty vector or SseK3-expressing plasmid. Cell lysates were harvested at indicated time points. Shown are immunoblotting using anti-Arg-GlcNAc and anti-tubulin

antibodies.

Q2. For Fig5, test Rab1 function and protein ER-to-Golgi transport using R-GlcNAcylation-deficient Rab1 mutants (Reviewer 1).

Corresponding to the question: In Figure 5, the authors performed a series of imaging and biochemical assays to conclude that Ssek3 inhibits Rab1 function (binding GDI-1/2, member-cytosol recycling). One most significant assay should be testing Rab1 function and protein ER-to-Golgi transport using R-GlcNAcylation-deficient Rab1 mutants (R4K or R4A) since they are available from the Figure 4. These assays will be complement to the assays using enzyme dead mutant of Ssek3 (Fig 5h, right panel).

For the ER-to-Golgi transport assay, there are some technical barriers. To avoid the function of the endogenous Rab1, we should transfect the Rab1 R4A or R4K mutant into *rab1a/b* double-knockout cell lines. Unfortunately, no *rab1a/b* knock-out cell lines are available because Rab1A/B are critical for cell survival and growth.

R72, R74, and R82 exist in the Switch-II region of Rab1, and R111 exists in the RabSF3 region, which is close to the Switch-II region in the tertiary structure. In Fig 4d and Fig 4e, R-GlcNAcylation-deficient Rab1 4A mutants have already shown a deficiency in GEF and GTPase activity. In addition, the critical role of the switch II region is well-defined. Crystal structures have shown that Arg74 and Arg82 participate in Rab-GDI binding^[1]. We do not think to test the function of R4A, and R4K mutants in this system adds too much to the field. The question that if the modification catalyzed by SseK3 inhibits Rab1 function is more meaningful, which have been answered in Fig5.

Q3. Clarify all points on Fig6 and address all points raised for the figures and transcript levels for IL-6 and TNF (Reviewer 1).

Q3.1: First, the ELISA data of IL-6 and TNF are less than 30 pg/ml in macrophage cell culture after salmonella stimulation for 6-18 h. It is hard to understand why the numbers are so low. What did it mean “...normalized to CFUs enumerated from each strain at 18 h post-infection”?

We adopted the reported method^[2] to calculate these values by normalizing ELISA data to CFUs enumerated from each strain after infection in order to exclude the difference due to the different intracellular bacterial numbers. The value of ELISA is divided by the number of bacteria, so the ELISA data of IL-6 and TNF are relatively low. Our statements in the manuscript were not clear, and we will improve this in the revised version.

Q3.2: Second, how does Ssek1 or 3 inhibits cytokine production in macrophages upon salmonella challenge? If it is related to signaling transduction, which target(s)? I do not think TNF signaling has something to do with salmonella-elicited cytokine production. It should be TLR signaling, for most of parts.

At the beginning of infection, TLR4 mediated NF- κ B activation is the major source of cytokine production. However, SseK1 and SseK3 are SPI-II effectors and play their roles not earlier than 6 h post-infection. One report showed that, during *Salmonella* infection, SseK1 could modify the mammalian signaling protein TRADD, and SseK3 could modify the death domains of receptors of the mammalian TNF superfamily, TNFR1, and TRAILR^[4]. So SseK1 and SseK3 have a TNF α -specific inhibitory effect on NF- κ B signaling and do not inhibit NF- κ B activation downstream of TLR4^[3]. Additionally, SseK1 and SseK3 were previously reported to inhibit the transcription of interleukin 6 (*il-6*) upon *Salmonella* challenge in *Tlr4*^{-/-} iBMM cells at 16 h post-infection^[3]. They also inhibit the NF- κ B-dependent luciferase gene transcription in RAW264.7 cells 16 h post-infection^[3]. Considering all the above together, TNF signaling may play a role in the late stage of *Salmonella* infection. However, we do not know it is through direct activation or feed-back loop activation.

Q3.3: Third, what is the transcript levels of IL-6 and TNF in Fig. 6a and b? The results may help to distinguish the scenarios of cytokine production versus cytokine release.

SseK1 and SseK3 are SPI-2 effectors. They transcribe, secrete, and catalyze PTM at the late stage of infection (Fig R2). The transcription level of IL-6 and NF- κ B reporter has been shown in previous study and stated in the former question Q3.2. To diminish the effects of the cytokine produced and secreted in the early stage, the culture medium was replaced with fresh medium at 6 h post-infection. The cytokine IL-6 (Fig. 6a) and TNF (Fig. 6b) secretion during 6-18 h after infection were measured by ELISA. It is hard to dissect the transcription inhibition and secretion inhibition in the bacterial infection system. To solve this problem, we used the “artificial IFN- γ system” and found that SseK3 but not SseK1 inhibit the protein secretion process.

Q3.4: Fourth, IFN- γ is a T cell cytokine, I do not feel it will add any significance for the physiological role of Ssek3 in cytokine secretion to artificially use a salmonella-T cell cytokine system.

In the context of Fig 6c, we did not intend to show the effects of *Salmonella* infection induced IFN- γ production. It was a biochemical assay, just like VSV-G transport (Fig 5h) and hGH release (Supplementary Fig 17), to show SseK3 can inhibit the secretory pathway, especially a cytokine secretion. Maybe we should validate this with more cytokines, such as IL-6, TNF- α , et al. However, considering we cannot do any experiments these months, we as to defer these experiments, thus question, to another time.

Q3.5: Finally, in Fig 6d, it is not a usual way to show intracellular growth of salmonella like replication fold. A growth curve will be helpful. Meanwhile, I am still not sure if any difference in intracellular bacterial growth is due to the cytokine difference. It is a unapproved assumption.

We agree with the referee that a growth curve will be helpful to show the dynamic intracellular replication of *Salmonella*. In Supplementary Fig 19, we showed a dynamic growth curve in HeLa cells. However, *Salmonella* replication in macrophages is highly dependent on SPI-2 effectors, and *Salmonella* begins to replicate in RAW264.7 cells at 10 h post-infection^[5]. The 2 h time point was used as measures of invasion, and the 24 h time point was used to calculate the replication index since it provided enough time for *Salmonella* to replicate. This method is widely used to show intracellular growth of *Salmonella* in other reports^[6-10], and we think it valid.

Additionally, we did not mean to set up the causal relationship between cytokine release and intracellular bacterial growth. They are two parallel observations. Thank you for your suggestion. It is insufficient to say, “SseK3 is crucial for pathogenesis by inhibiting cytokine production and secretion”, and we will edit the text to make it clear.

Q4. Include proper controls for the different cell lines and provide a rationale for using these cell lines (Reviewer 2).

Corresponding to the comments from reviewer 2: the authors infect transformed epithelial cellular models containing with sun-tag modified SseK proteins to imitate previously published infectious models on macrophages. However, the provided data was not obtained from a single cell line, HeLa and 293T cells were used to complete different experimental and mixing data from two immortalized cells lines could lead to the wrong interpretations. If two cell lines are to be used, proper controls to assure that the observed phenotypes are conserved in both cellular systems are required to validate the presented data. It is also not clear why these non-physiological systems were used if primary cells/more appropriate cell lines are available.

We showed the localization of SseK proteins in HeLa cells with the SunTag system for two reasons. 1) We can observe the dynamic process to gain the time window of specific localization; 2) Showing this subcellular localization of SseK3 is not macrophage-specific, so the immortalized cell lines could be used for further research. We knew that the scientific advance of these data was not too much, so we arranged the majority of these SunTag data in supplementary materials.

In our manuscript, 293T and HeLa cells were used to complete different assays. Because of the high transfection efficiency and low cytotoxicity, 293T cells were used to do transfection and immunoprecipitation experiments. HeLa cells were widely used in immunofluorescence experiments because of their clear and obvious organelles.

During our investigation, we did most of the assays in both cell lines. Overexpression of SseK3 destroyed the structure of Golgi (Fig R4) and inhibited ER to Golgi trafficking (Fig R5). The phenotypes were conserved in both cell lines. These figures will be further arranged and added to supplementary data.

Fig R4 Overexpression of SseK3 destroys the structure of Golgi both in 293T and HeLa cells. Effects of SseK3 transfection on endogenous Golgi protein immunostaining in 293T (left) or HeLa cells (right). 293T or HeLa cells were transfected with a plasmid expressing RFP/GFP alone, RFP/GFP-tagged-SseK3 or its dead mutant. Shown are immunofluorescence staining using an anti-ManII antibody.

Fig R5 SseK3 blocks VSV-G trafficking from Golgi to the plasma membrane in both 293T and HeLa cells. VSV-G-GFP-expressing 293T (left) or HeLa cells (right) transfected with plasmids expressing indicated RFP-tagged-proteins were incubated at 40.5 °C for 16 h and then moved to 32 °C for 4 h. Shown are confocal fluorescence images of VSV-G-GFP localization.

Q5. Include inactive SseK1,3 controls (Reviewer 2).

Corresponding to the comment: In the Supplementary figure 1b the only control used is an empty vector. However, the inactive SseK1,3 controls are missing. This is important as the used sun tag plus the 24 GFP molecules weigh more than 1000KDa and this could lead to migration artefacts.

We are grateful for this advice. In the Supplementary figure 1b, we studied the localization of Arginine-GlcNAcylated proteins catalyzed by the SseK effectors during infection. This assay was applied in HeLa cells but not HeLa-scFv-GFP cells. So the localization of Arg-GlcNAcylated proteins would not be effected by 24*GFP molecules. We added the inactive SseK3 DxD as a control in Fig R6. To exclude the migration artifacts due to SunTag system and to supplement with Supplementary figure 1a, we provide the inactive SseK1,3 controls in Fig R7. These figures will be further arranged and added to supplementary data.

Fig R6 Subcellular localization of Arg-GlcNAcylated proteins catalyzed by the SseK effectors during *Salmonella* infection. HeLa cells were infected with *S. Typhimurium* $\Delta sseK1/2/3$ complemented with empty vector, SseK1- or SseK3-expressing plasmid as indicated. Shown are immunofluorescence detection of Arg-GlcNAc (red) and Golgi apparatus (green). Data are representative of three independent experiments.

Fig R7 Golgi-localization of T3SS-translocated SseK1 and SseK3. Subcellular localization of T3SS-translocated SseK1 and SseK3. HeLa cells stably expressing scFv-GCN4-GFP were infected with *S. Typhimurium* Δ sseK1/2/3 complemented with a plasmid expressing indicated SunTag₂₄-tagged proteins. Mock (no infection) cells were set as a negative control. Shown are fluorescence images taken at 14 h post-infection.

Q6. Address the point related to data not fully supporting “PtdIns(4,5)P2 is essential for SseK3 recruitment to the Golgi” brought up by Reviewer 2 and address how the statement can be further validated (Reviewer 2).

Corresponding to the comment: To explain the observed location of SseK3 in the Golgi apparatus the authors identified an electronegative patch constituted by the residues K87/R89/K234, the latter two conserved in the NleB1 orthologue which doesn't migrate to the Golgi. Figure 1e shows a reduced but not abolished binding to the PtdIns(4,5)P2 for the K87A/R89A/K234A mutant. However, presented data shows strong binding to PI3P, PI5P and probably to PI(3,5)P2 regardless of the presence of the K87A/R89A/K234A mutations. This, together with the ability of sseK3 to, still, being recruited to the Golgi under rapamycin treatment suggest other interactions outside the proposed electronegative patch (or other protein factors) and supports a non-disclosed recruitment path that might work independently of the PtdIns(4,5)P2 presence.

Suggesting that, in opposition to authors claim, PtdIns(4,5)P₂ is not essential for SseK3 recruitment to the Golgi. This requires further experimental clarification.

Thank you for these questions. We compare the electrostatic potential surface between SseK3 and NleB as shown in Fig R8. The Lys87, Arg89, and Lys234 in SseK3 can form a polybasic patch, while the Asn84, Arg86, and Lys229 in NleB cannot. Although NleB and SseK3 harbor two conserved residues in this region, they are quite different in surface charge properties.

The K87A/R89A /K234A mutant lost the Golgi localization and showed a significant decrease in binding affinity for PI(4,5)P₂. First, both the WT SseK3 and the K87A/R89A/K234A mutant showed a strong binding affinity to PI(3)P and PI(5)P, whereas the K87A/R89A /K234A lost the localization. This phenomenon indicated that binding to PI(3)P and PI(5)P was not responsible or not sufficient for Golgi localization. Second, this localization-deficient mutant exhibited an apparently lower binding affinity to PI(4,5)P₂. These data indicated one hypothesis that binding to PI(4,5)P₂ might contribute to the Golgi localization of SseK3. To further prove this hypothesis, we applied the inducible recruitment system of phospholipid phosphatase. PIP4P1 is a PI(4,5)P₂ specific phosphatase. Co-expression of SseK3 and PIP4P1 efficiently decreased the Golgi-localization of SseK3. These differences shown in Fig. 1f are significant statistically. So in the manuscript, we stated that binding to PI(4,5)P₂ was crucial for the Golgi-localization of SseK3.

Meanwhile, the same with the reviewer's consideration, we do not exclude other possibilities that may contribute to the SseK3 Golgi-localization.

First, as shown above in Fig R7, SseK3-DxD mutant needed an additional 6 h to localize on Golgi structure with the same extent to wild-type SseK3. According to our research, SseK3 modified Rab small GTPases, disturbed the membrane trafficking cycle, and trapped the VSV-G protein in the Golgi apparatus. Hence, the enzymatic activity of SseK3 may contribute a little bit to the Golgi localization of itself.

Second, we asked if SseK3 binds with Golgi-related protein to show the Golgi localization. We tried hard to search for this assumed binding partner. Unfortunately, we have not found a Golgi-localized binding protein through IP-MASS and yeast two-hybrid screen until now.

According to the above points, we conclude that binding with PI(4,5)P₂ is crucial for the Golgi-localization of SseK3, the modification on Rab GTPases also has a little bit contribution. We can discuss more in the revised manuscript.

Fig R8. Electrostatic potential surface schemes of Ssek3 (left, PDB code:6EYT) and NleB (right, PDB code:6ACI). The coloration from red to blue represents negatively to positively charged regions. The polybasic patches are circled with the residues K87/R89/K234 in Ssek3 and N84/R86/K229 in NleB shown in sticks. All structure figures were prepared in PyMOL and the surface potential were calculated using the APBS electrostatics tools with default parameters. The corresponding positive patch in NleB becomes smaller after the K87 in Ssek3 is replaced by N86 in NleB.

Q7. Include the electrostatic potential scale and the method for calculating some values in S3 (Reviewer 2).

Corresponding to the question: In supplementary figure 3a, the electrostatic potential scale and the method for calculating the values are missing.

We are grateful for this advice. We will add these details in the revised manuscript as shown in Fig R9.

Fig R9: Electrostatic potential surface schemes of Ssek3 (PDB code:6EYT) from different perspectives. The coloration from red to blue represents negatively to positively charged regions. The polybasic patches are circled with the basic amino acid residues shown in sticks. Adjacent schemes are changed by rotating the indicated angle along X (horizontal line) or Y (vertical line) axis. All structure figures were prepared in PyMOL and the surface potential were calculated using the APBS electrostatics tools with default parameters.

Q8. Determine the presence/absence of additional PTM in the different backgrounds to validate the conclusion that Rab1 prenylation is crucial for arginine GlcNAcylation by SseK3 (Reviewer 2).

Corresponding to the question: The authors state that Rab1 prenylation is important for the arginine modification but the presented experimental setup can only prove that silencing the Rab1 prenylation signal (CC) abolishes trafficking to the Golgi. The authors imply a direct effect but this might as well be indirect (i.e. involving other prenylated protein partners). Furthermore, Rab1 could potentially be modified by additional modifications (PTM) which makes it a target for SseK3. To clarify this issue intact mass spec to determine the presence/absence of additional PTM in the Δ CC and the CC-SS vs the WT is required.

In Fig 3 we show that, 1) SseK3 exhibits a markedly higher enzymatic activity towards Rab1 in 293T cells than that in *E.coli*. 2) CC-SS or Δ CC mutants of Rab1 cannot be modified by SseK3 neither in ectopic transfection nor in *Salmonella* infection. 3) 293T cell-purified but not *E.coli*-purified-Rab1 can be modified by SseK3 in reconstitution system. 4) Prenylation is also important for the modification on Rab8 and Rab33 catalyzed by SseK3. We do not know the effect of prenylation is direct or indirect. Thank you for this enlightening question. To answer this question, a proper experimental design would be using *in vitro* system. Bacterially purified Rab1, adding geranylgeranyl-group by RabGGTase, and then subject to Arg-GlcNAcylation reaction *in vitro*.

We also curious about if there are any additional PTMs on the CC-SS or Δ CC mutants of Rab1. If there are, do they have effects on the arginine GlcNAcylation catalyzed by SseK3? However, due to the new coronavirus outbreak, we cannot do these experiments for a long time as stated above. No matter the answer is yes or no, it has no effects on our conclusion that Rab1 prenylation is crucial for arginine GlcNAcylation by SseK3. May I ask for an opportunity to show this detail information in future work?

Q9. Provide *in vitro* GlcNAc transfer activity assay onto Rab1 peptides to validate the modification sites (Reviewer 2).

Corresponding to the question: The authors only use mass spectrometry data and site point SseK3 mutants to search and validate the putative Rab1 modification sites. Arg72, Arg74, Arg82 and Arg111 are proposed based on their experimental data. However, the structural architecture of Rab1 makes it highly improbable that these sites are modified.

Further data, like *in vitro* GlcNAc transfer activity assay onto Rab1 based peptides is required to validate this observation, especially since the quadruple R4K mutant referred in Supplemental Figure 4b or the mutants analyzed in Supplemental Figures 4d and 4e does not provide clarification which is/are the real Rab1 glycosylation sites.

We use mass spectrometry and Arg-GlcNAc specific antibody detection on Rab1 mutants to search and validate the modification sites. We believe they are widely used approaches to detect and validate post translational modifications.

It is well-known that the *O*-GlcNAc transferase (OGT), which can trigger GlcNAc modification on Ser/Thr, can modify peptides derived from its substrate proteins^[11]. In our previous study, we found that NleB, the orthologue of SseK proteins, did not modify any peptides derived from its targets, TRADD, and FADD. Accordingly, a C-shape binding cave is required for substrate recognition shown in the NleB/FADD-DD complex structure^[12]. So SseK3 may not modify Rab1 peptides. Even if we can detect some modifications on the peptide, it is not clear whether the forced modifications of a peptide *in vitro* is of any physiological relevance. From our data, R72, R74, R82, and R111 are all modification sites.

Thanks for your suggestion. Our following project would be to reveal the SseK3/Rab1 complex structure, which may provide the structural and functional insights to the interaction of Rab1 and SseK3.

Q10. Correct the text reference to figures and correct potential mistakes in the text (Reviewer 2).

Corresponding to the question: the authors get confused and referred to main text figures while the data is instead shown in supplemental files. (lines 257 to 262). This and many other errors in the text makes it very hard to understand their findings. Text and figures often mention the vague term “mammalian cells”, which is not helpful for the reader. Would be better to indicate which specific cell lines were used instead.

Thank you for the advice. We will fix these problems in the revised manuscript.

Q11. Provide scatter plots for Fig5a and b (Reviewer 2).

Corresponding to the question: Figure 5a and 5b. Bar representations are not appropriate to assess the quality of the data, scatter plots are required instead.

Thank you for your suggestion. The figure shown in Fig R10 will be arranged and added to the revised manuscript.

Fig R10. Effects of arginine GlcNAcylation of Rab1 on the GTPase activity and GEF activity. Recombinant Rab1 and modified Rab1 were subjected to GTPase activity assay with the addition of GAP protein LepB or subjected to GEF assay with the addition of GEF protein DrrA. The percentages of relative activity are mean \pm SD from three independent experiments.

Q12. Improve the readability of the manuscript perhaps by taking advantage of an editorial service.

Our manuscript will be edited by a native English speaker and revised to improve readability.

Q13. Address all clarifications in the text/figures/methods.

We will do that as your suggestions.

Another four comments/questions from reviewers not mentioned above are listed and answered below.

Q14 (from Reviewer 1). In Figure 4 b, d and e, the authors showed the loss of R-GlcNAcylation of Rab1 when four R residues were mutated to lysine and the loss of GTPase activity/GEF activity when four R residues were mutated to alanine. What is the reason to perform assays using different mutations? The reviewer is surprised that both GTPase activity and GEF activity of Rab1 were defective when R-GlcNAcylation was removed, which indicates an unspecific inhibitory effect of Ssek3-mediated R-GlcNAcylation on Rab1 function. This phenomenon is different from other effector salmonella effectors (references 31-33). Could the authors add some discussion on this issue?

We mutated four arginine residues to four alanine residues or four lysine residues. We showed the data that R4K could not be modified in Fig 4b. Below in the Fig R11, the data show that R4A mutant cannot be modified, either.

In the previous manuscript, we listed some *Salmonella* effectors that have been reported to modulate the activity of host GTPases by mimicking host regulators, like

GAP or GEF (references 31-33). However, SseK3 is not a GEF or GAP. SseK3 catalyzes post translational modifications on Rab GTPases in the switch-II region. The switch-II region is critical for Rab GTPase activity and GEF activity. Similar to SseK3, bacterial effectors from *Legionella* disturb Rab1 function by covalently modifying at residues within switch II region. The AnkX phosphorylates Rab1 at Ser76 and affects its activity in GTP loading stimulated by SidM (GEF) and hydrolysis induced by LepB (GAP)^[13]. AMPylation on Tyr77 by DrrA disrupts the ability of Rab1 to mediate vesicle transport within the secretory pathway in eukaryotic cells^[14-15]. SetA glucosylates Rab1 at Thr75, thus attenuates its GTPase activity and inhibiting its interaction with the GDI^[16]. We will add this discussion in the revised manuscript as the reviewer's kindly suggestion.

Fig R11. In vivo modification of Rab1 arginine mutants by SseK3 during Salmonella infection. 293T cells were transfected with the WT and mutated Flag-tagged Rab1, and then subjected to pathogen infection. Shown are immunoblots. Representative data from at least three independent experiments are shown.

Q15 (from Reviewer 2). Figure 3a is not clear. The second lane shows GlcNAcylated Rab1 but no signal from Rab1, same for the third lane. Several lanes are shown increased signal for GlcNAcylated Rab1 but no explanation is given for these effects.

In Fig 3a, all the Rab1 proteins in this assay were co-expressed with SseK3 in 293T or in *E. coli*. Lane 1 was loaded with Rab1 purified from *E. coli*. While lane 2-7 were loaded with Rab1 purified from 293T in gradient loading. 1) When lane 1 and lane 2 are compared, the Rab1 protein in lane 2 are much less than in lane 1, whereas the Arg-GlcNAcylated-Rab1 are the same with lane 1. 2) Lane 1 and 7 are loaded with the same amount of Rab1, whereas lane 7 shows much higher signal when detected by the Arg-GlcNAc antibody. The data show that the SseK3 exhibits a markedly higher enzymatic activity towards Rab1 in 293T cells than that in *E. coli*. To improve the readability, we have added an indicator icon as shown in Fig R12, and this figure will be further arranged and added into the revised manuscript.

Fig R12. Arg-GlcNAcylation detection of Rab1 purified from prokaryotic and eukaryotic cells. Anti-Rab1 is shown as loading control.

Q16 (from Reviewer 2). In Figure 3d, would be useful to indicate which sample was recombinantly produced instead of expecting the reader to guess it by the Rab1 molecular size shift.

Thank you for the suggestion. To improve the readability, we will fix these problems as suggested in the revised manuscript.

Q17 (from Reviewer 2). The RAW264.7 cell line is used to check for the replication of WT *S. Typhimurium* and Δ sseK1/2/3 mutant. However, RAW264.7 is a murine cell line and human counterpart/primary cells would be useful as a validation control.

Salmonella exhibits quite differences in survival, replication, and apoptosis-inducing ability within human and murine macrophages. *S. Typhimurium* replicates in RAW264.7 cells at 10 h.p.i., but poorly in primary human macrophages because of their divergent repertoires of the virulence factors that responsible for the pathogenesis^[17-18]. The SPI-2 T3SS is involved in intracellular survival of *S. Typhimurium* and systemic disease. In contrast, the intracellular survival of *S. Typhi* in human macrophages is independent of *Salmonella* pathogenicity island (SPI)-2^[19]. So the SPI-2 T3SS effectors may not be involved in the replication of *S. Typhimurium* in human macrophages.

Compared to other murine macrophages (such as, murine bone marrow-derived macrophages, BMDM), *S. Typhimurium* induces a lower degree of cytotoxicity to RAW264.7 and replicates to a higher fold^[20]. Thus, it is a common assay to study effects of bacterial effectors on the *S. Typhimurium* replication^[21].

References

1. Rak A, Pylypenko O, Durek T, et al. Structure of Rab GDP-dissociation inhibitor in complex with prenylated YPT1 GTPase[J]. *Science*, 2003, 302(5645): 646-650.
2. Pilar A V C, Reid-Yu S A, Cooper C A, et al. GogB is an anti-inflammatory effector that limits tissue damage during Salmonella infection through interaction with human FBXO22 and Skp1[J]. *PLoS pathogens*, 2012, 8(6).
3. Günster R A, Matthews S A, Holden D W, et al. SseK1 and SseK3 type III secretion system effectors inhibit NF- κ B signaling and necroptotic cell death in Salmonella-infected macrophages[J]. *Infection and immunity*, 2017, 85(3): e00010-17.
4. Newson J P M, Scott N E, Chung I Y W, et al. Salmonella effectors SseK1 and SseK3 target death domain proteins in the TNF and TRAIL signaling pathways[J]. *Molecular & Cellular Proteomics*, 2019, 18(6): 1138-1156.
5. Monack D M, Raupach B, Hromockyj A E, et al. Salmonella typhimurium invasion induces apoptosis in infected macrophages[J]. *Proceedings of the National Academy of Sciences*, 1996, 93(18): 9833-9838
6. Brumell J H, Rosenberger C M, Gotto G T, et al. SifA permits survival and replication of Salmonella typhimurium in murine macrophages[J]. *Cellular microbiology*, 2001, 3(2): 75-84.
7. Lee A K, Detweiler C S, Falkow S. OmpR regulates the two-component system SsrA-SsrB in Salmonella pathogenicity island 2[J]. *Journal of bacteriology*, 2000, 182(3): 771-781.
8. Govoni G, Canonne-Hergaux F, Pfeifer C G, et al. Functional expression of Nramp1 in vitro in the murine macrophage line RAW264. 7[J]. *Infection and immunity*, 1999, 67(5): 2225-2232.
9. Drecktrah D, Knodler L A, Galbraith K, et al. The Salmonella SPI1 effector SopB stimulates nitric oxide production long after invasion[J]. *Cellular microbiology*, 2005, 7(1): 105-113.
10. Buckner M M C, Croxen M, Arena E T, et al. A comprehensive study of the contribution of Salmonella enterica serovar Typhimurium SPI2 effectors to bacterial colonization, survival, and replication in typhoid fever, macrophage, and epithelial cell infection models[J]. *Virulence*, 2011, 2(3): 208-216.
11. Smet-Nocca C, Broncel M, Wieruszkeski J M, et al. Identification of O-GlcNAc sites within peptides of the Tau protein and their impact on phosphorylation[J]. *Molecular BioSystems*, 2011, 7(5): 1420-1429.
12. Ding J, Pan X, Du L, et al. Structural and functional insights into host death domains inactivation by the bacterial arginine GlcNAcyltransferase effector[J]. *Molecular cell*, 2019, 74(5): 922-935. e6.
13. Tan Y, Arnold R J, Luo Z Q. Legionella pneumophila regulates the small GTPase Rab1 activity by reversible phosphorylcholation[J]. *Proceedings of the National Academy of Sciences*, 2011, 108(52): 21212-21217.

14. Müller MP, Peters H, Blumer J, Blankenfeldt W, Goody RS, Itzen A. The *Legionella* effector protein DrrA AMPylates the membrane traffic regulator Rab1b. *Science* 2010; 329:946 - 9;
15. Neunuebel MR, Chen Y, Gaspar AH, Backlund PS Jr., Yergey A, Machner MP. De-AMPylation of the small GTPase Rab1 by the pathogen *Legionella pneumophila*. *Science* 2011; 333:453 - 6;
16. Wang Z, McCloskey A, Cheng S, et al. Regulation of the small GTPase Rab1 function by a bacterial glucosyltransferase[J]. *Cell discovery*, 2018, 4(1): 1-13.
17. Schwan W R, Huang X Z, Hu L, et al. Differential bacterial survival, replication, and apoptosis-inducing ability of *Salmonella* serovars within human and murine macrophages[J]. *Infection and immunity*, 2000, 68(3): 1005-1013.
18. Forest C G, Ferraro E, Sabbagh S C, et al. Intracellular survival of *Salmonella enterica* serovar Typhi in human macrophages is independent of *Salmonella* pathogenicity island (SPI)-2[J]. *Microbiology*, 2010, 156(12): 3689-3698.
19. Sabbagh S C, Forest C G, Lepage C, et al. So similar, yet so different: uncovering distinctive features in the genomes of *Salmonella enterica* serovars Typhimurium and Typhi[J]. *FEMS*.
20. Monack D M, Raupach B, Hromockyj A E, et al. *Salmonella typhimurium* invasion induces apoptosis in infected macrophages[J]. *Proceedings of the National Academy of Sciences*, 1996, 93(18): 9833-9838.
21. Buckner M M C, Croxen M, Arena E T, et al. A comprehensive study of the contribution of *Salmonella enterica* serovar Typhimurium SPI2 effectors to bacterial colonization, survival, and replication in typhoid fever, macrophage, and epithelial cell infection models[J]. *Virulence*, 2011, 2(3): 208-216.

REVIEWERS' COMMENTS:

Reviewer #1 (Remarks to the Author):

The revision has successfully addressed the Reviewer's concerns.

Reviewer #2 (Remarks to the Author):

Due to the worldwide COVID-19 lockdown situation, which we all are suffering, the authors have attempted to address the issues that were identified in the first submission. However, several issues remain unresolved, with changes being largely cosmetic.

Strengths:

The authors have included convincing evidence of the SunTag suitability in their particular experimental setup and evidence of similar phenotypes in the cell lines used. Control data for Inactive SseK3 has been provided as suggested. Scales and method for electrostatic calculation have been provided as requested. Figure references and other mistakes are now corrected as requested. Scatter plots are now shown as requested.

Weaknesses:

- 1) The authors discuss the differences in the overall electrostatic surface potential between SseK3 and NleB1. However, the main control experiments to validate the importance of the electropositive Lys87, Arg89 and Lys334 remain undone. The authors should really include point mutants of each residue, in particular the Lys87 to Asn to generate a NleB-like SseK3.
- 2) The issue with prenylation remains unresolved, with the authors claiming they are currently not able to access the equipment. If they want to keep the statement that prenylation is required for SseK3 GlcNAcylation on several Rab proteins, then they need to provide experimental evidence – perhaps after the lockdown.
- 3) There remains a concern about the proximity of the modification sites and the issue of avoiding false-positive results in mass spectrometry data. This needs to be resolved with experimental data, again to be provided after the lockdown.

Reviewer #3 (Remarks to the Author):

Overall, I found the MS to be an impactful paper with several interesting conclusions supported by many lines of evidence.

Although the authors do not have direct experimental evidence that the recombinant Rab is prenylated, the authors ruled out the involvement of other prenylated protein partners in relation to the Rab modification by SseK3 using Ecoli- and eukaryotic cell-derived Rab. However, it is possible that other eukaryotic cell-dependent modifications of Rab1 are required beyond prenylation, but this can, and should, be addressed in the discussion.

I suggest the authors to expand their discussion to include acknowledgment of the weaknesses